

# Spatio-temporal variability of Arctic summer temperatures over the past two millennia: an overview of the last major climate anomalies

Johannes P. Werner[1], Dmitry V. Divine[2,3], Fredrik Charpentier Ljungqvist[4,5], Tine Nilsen[3], and Pierre Francus[6,7]

[1]Bjerknes Centre for Climate Research and Department for Earth Science, University of Bergen, PO Box 7803, N-5020 Bergen, Norway
[2]Norwegian Polar Institute, FRAM Centre, N-9296 Tromsø, Norway.
[3]Department of Mathematics and Statistics, University of Tromsø – The Arctic University of Norway, N-9037, Norway
[4]Department of History, Stockholm University, SE-106 91 Stockholm, Sweden
[5]Bolin Centre for Climate Research, Stockholm University, SE-106 91 Stockholm, Sweden
[6]Institut National de la Recherche Scientifique Centre – Eau Terre Environnement 490 rue de la couronne, Québec, QC G1K 9A9, Canada
[7]GEOTOP Research Center, Montréal, Canada

*Correspondence to:* J.P. Werner (johannes.werner@geo.uib.no)

**Abstract.**

In this article, the first spatially resolved millennium-long summer (June–August) temperature reconstruction over the Arctic and Subarctic domain (north of 60° N) is presented. It is based on a set of 54 annually dated temperature sensitive proxy archives of various types, mainly from the updated and revised PAGES2k database supplemented with 6 new recently published proxy records. As a major novelty, an extension of the Bayesian BARCAST climate field (CF) reconstruction technique provides a means to treat climate archives with dating uncertainties. In total 1400 realisations of the temperature CF were generated, enabling further analyses to be carried out in a probabilistic framework. The new seasonal CF reconstruction for the Arctic region can be shown to be skilful for the majority of the terrestrial nodes. The decrease in the proxy data density back in time however limits the analyses in the spatial domain to the period after 750 CE, while the spatially averaged reconstruction covers the entire time interval of 1–2002 CE. The analysis of basic features of the reconstructed seasonal CF focuses on the regional expression of past major climate anomalies in order to uncover the potential of the new product for studying Common Era temperature variability in the region.

The long-term, centennial to millennial, evolution of the reconstructed temperature is in good agreement with a general pattern that was inferred in recent studies for the Arctic and its sub-regions. On the pan-Arctic scale the reconstruction shows a cooling trend which is, however, statistically insignificant and the estimated magnitude of the millennial scale cooling is three times smaller than inferred in the previous studies. The trend is spatially heterogeneous and for some regions such as Greenland the reconstruction demonstrates a tendency to the warming instead. The reconstruction shows a pronounced Medieval Climate Anomaly (MCA, here, ca. 960–1060 CE), which was characterised by a sequence of extremely warm decades over the whole domain. The medieval warming was followed by a gradual cooling into the Little Ice Age (LIA), with 1580–1680 CE as the longest centennial-scale cold period, culminating around 1812–1822 CE for most of the target region. At the same time there is





evidence for a drastic reduction in sea-ice on the Greenland shelf, which is reflected by rather high summer temperatures over Greenland and Baffin Island during that decade.

During the MCA, the contrast between reconstructed summer temperatures over mid- and high-latitudes in Europe and the European/North Atlantic sector of the Arctic shows a very dynamic expression of the Arctic amplification, with leads and lags between continental and more marine and extreme latitude settings. While our analysis shows that the peak MCA summer temperatures were as high as in the late 20$^{th}$ and early 21$^{st}$ century, the spatial coherence of extreme years over the last decades seems unprecedented at least back until 750 CE. However, statistical testing could not provide conclusive support of the contemporary warming to supersede the peak of the MCA in terms of the pan-Arctic mean summer temperatures.

## 1  Introduction

During the past decades, the Arctic has experienced a more rapid and pronounced temperature increase than most other parts of the world. The dramatically shrinking extent of Arctic sea-ice in recent years – with a decline in both minimum extent in summer and maximum area in winter – accompanied by a transition to a younger and thinner sea ice cover, is often interpreted as the clearest and most unambiguous evidence of anthropogenic global warming (Comiso et al., 2008; Perovich et al., 2008; Serreze et al., 2007; Maslanik et al., 2011; Meier et al., 2014). Additionally, the Arctic region is of utmost importance in the context of global climate and global climate change. Reduction in perennial sea ice cover leads to increased heat transport northward (Müller et al., 2012; Smedsrud et al., 2008), as well as changes the Arctic energy balance due to positive albedo feedbacks (Curry et al., 1995; Miller et al., 2010; Perovich et al., 2002, 2011). Melting of permafrost can release methane (CH$_4$), a more efficient greenhouse gas than carbon dioxide (CO$_2$), and likewise gives a positive feedback that may further amplify the temperature increase (O'Connor et al., 2010; Shakhova et al., 2010). Even partial melting of the Greenland inland ice cap and/or the numerous smaller high-latitude glaciers would significantly raise the global sea level and threaten to flood low-laying coastal regions around the world (Grinsted et al., 2010; Vermeer and Rahmstorf, 2009).

The instrumental temperature record is too short and spatially sparse to assess whether this recent warming, and the accompanying sea-ice reduction, experienced in the Arctic region so far, fall outside the range of natural variability on centennial to millennial time-scales. Moreover, general circulation models have limited capabilities in reliably simulating Arctic climate change on centennial time-scale and beyond (IPCC, 2013). The simplified parametrisations of dynamic and thermodynamic sea-ice processes, and the limited skills in describing ocean–sea-ice–atmosphere energy exchange, in particular in modelling polar clouds and oceanic heat flux, is especially evident from the lack of skill in reproducing the present-day rapid loss of Arctic sea-ice (e.g. Hunke et al., 2010). Hence both the possible uniqueness during the Common Era (CE) of the on-going Arctic warming and the relative role of anthropogenic and natural forcings driving the process are difficult to fully assess without a longer perspective from palaeoclimate proxy-based temperature reconstructions. Thus palaeoclimate data that can be used for understanding the range of natural climate variability in the Arctic region over long time-scales are needed, and an effort needs to be made to integrate different types of information from a variety of palaeoclimate archives.





Especially the spatial signature of past Arctic temperature variability remains poorly understood and has recently been a contested issue. Young et al. (2015), based on results from relatively poorly age-constrained moraine dates of glacier advances,
questioned the long-standing notion within palaeoclimatology that Greenland was exceptionally warm around the time of Norse settlement (in the 980s CE). They argue that the medieval warming in the Arctic did not extend to Greenland, thus challenging our long-held understanding of the settlement and later abandonment of Norse Greenland. This is in direct contradiction to the spatial temperature reconstructions of Mann et al. (2009) and Ljungqvist et al. (2012, 2016), which instead point to very warm conditions in Greenland at that time. As we will see, our results show that the Greenland sector of the Arctic indeed was warm
during the 10th and especially the early 11th centuries CE.

Since the 1990s, several multi-proxy reconstructions of Arctic and Subarctic (usually 90–60° N) temperatures have been published. The first one of those was the multi-proxy reconstruction by Overpeck et al. (1997), who compiled 29 proxy records from lake sediment, tree-ring, glacier, and marine sediment records to present a decadally resolved uncalibrated index of temperature variability since 1600 CE. They found that the highest temperatures in the Arctic region since 1600 CE occurred after
1920 CE. Kaufman et al. (2009) published the first quantitative multi-proxy reconstruction of summer temperature variability in the Arctic (90–60° N) during the past 2,000-year at decadal resolution using the composite-plus-scaling method. This study concluded that the 20th century warming reverses a long-term orbitally driven summer cooling and that the mid- and late 20th century temperatures were the highest in the past two millennia.

Shi et al. (2012) published the first annually resolved multi-proxy summer (June–August) temperature reconstruction for
the Arctic region, extending back to 600 CE, based on a set of 22 proxy records with annual resolution. They utilised a novel ensemble reconstruction method that combined the traditional composite-plus-scale method – known to underestimate low-frequency variability (e.g. von Storch et al., 2004) – and the LOC method of Christiansen (2011) that exaggerates the high-frequency variability (c.f. e.g. Christiansen and Ljungqvist, 2017). The reconstructed amplitude of the centennial-scale summer temperature variability was rather dampened and found to be less than 0.5°C but with large year-to-year and decadal-
to-decadal variability. Shi et al. (2012) found a clear cold anomaly 630 to 770 CE, a peak warming ca. 950 to 1050 CE, and overall relatively cold conditions ca. 1200–1900 CE. However, three distinctly warmer periods during the Little Ice Age were reconstructed ca. 1470–1510, 1550–1570, and 1750–1770 CE. Contrary to Kaufman et al. (2009), Shi et al. (2012) found peak medieval Arctic summer temperatures in the 10th century to been approximately equal to recent Arctic summer temperatures.

Tingley and Huybers (2013) used BARCAST (Bayesian Algorithm for Reconstructing Climate Anomalies in Space and
Time Tingley and Huybers, 2010a), a method based on Bayesian inference of hierarchical models (see also sec. 3), to reconstruct surface-air temperatures of the last 600 years over land north of 60° N. The reconstruction is mostly based on the proxy dataset collected by the PAGES 2k Consortium (2013). They found that while the recent decades were the warmest over the last 600 years, the actual inter-annual variability has remained effectively constant. Much of the data used therein is common with the work presented here, with a few updated records (see section 2.2, and PAGES 2k Consortium, 2017).
Hanhijärvi et al. (2013) presented a 2000-year long annual mean temperature reconstruction for the North Atlantic sector of the Arctic (north of 60° N and between 50° W and 30° E) using 27 proxy records of various types, resolution and length employing the novel Pairwise Comparison (PaiCo) method. Their reconstruction reveals centennial-scale temperature varia-





tions of an amplitude of over 1°C, with a distinct Roman Warm Period, warm Medieval Climate Anomaly and 20th century warming. A somewhat indistinct Dark Age Cold Period is found in the middle of the first millennium CE, whereas a very

clear and persistently cold Little Ice Age extends from the mid-13th century until the turn of the 20th century, with the lowest temperatures in the 19th century. Peak temperatures during the Roman Warm Period and the Medieval Climate Anomaly were found to equal recent temperatures in the the North Atlantic sector of the Arctic. The PAGES 2k Consortium (2013) extended the PaiCo reconstruction to cover the whole Arctic (60–90° N), using 67 proxy records of various types, resolution and length to reconstruct annual mean temperature variations over the past two millennia. They reconstructed a generally rel-

atively warm first millennium CE, and a relatively indistinct Medieval Climate Anomaly, and a relatively cold Little Ice Age from ca. 1250 CE to 1900 CE. The amplitude of the reconstructed low-frequency temperature variability in the whole Arctic by the PAGES 2k Consortium (2013) is smaller than that reconstructed for only the North Atlantic sector of the Arctic by Hanhijärvi et al. (2013). A revised Arctic2k reconstruction was subsequently published by McKay and Kaufmann (2014), using an updated and corrected proxy database containing 59 records, showing a larger long-term cooling trend and is on average

ca. 0.5°C warmer prior to ca. 1250 CE than in PAGES 2k Consortium (2013). Peak temperatures during the Roman Warm Period and the Medieval Climate Anomaly thus approximately equal recent temperatures in McKay and Kaufmann (2014) as in Shi et al. (2012) and Hanhijärvi et al. (2013), instead of being much lower as in the Arctic2k reconstruction by the PAGES 2k Consortium (2013).

This study is mostly comparable with that of Tingley and Huybers (2013): our method is an update of theirs (Tingley

and Huybers, 2010a; Werner and Tingley, 2015), and the proxy network is an update of the PAGES2k database (PAGES 2k Consortium, 2017). There are a few notable differences: i) the CF reconstruction is performed on an equal area grid (land only), which should be more suitable for a spatially homogeneous process – especially at high latitudes. This target gridded instrumental dataset is directly derived from available data from meteorological stations. ii) The gridded reconstruction goes back into the first millennium CE. iii) The proxy dataset is larger and more extensively screened, and iv) the age uncertainties

of the proxies used are respected. Thus, the propagation of uncertainties from proxy data to the final reconstruction product is more complete. v) Additionally, while Tingley and Huybers (2013) use a single set of parameters for all proxies of one type, these are estimated here for each individual record. This potentially removes spurious precision at proxy sites responding less strong to the seasonal temperature anomalies, and should increase the precision at locations with stronger climate response.

## 2   Instrumental data and proxy data

The following section provides a short overview of the used instrumental and palaeoclimate proxy data. The quality of the input data and their distribution in space and time play a strong role in the reconstruction process and for the reconstruction reliability (c.f. e.g. Wang et al., 2015).





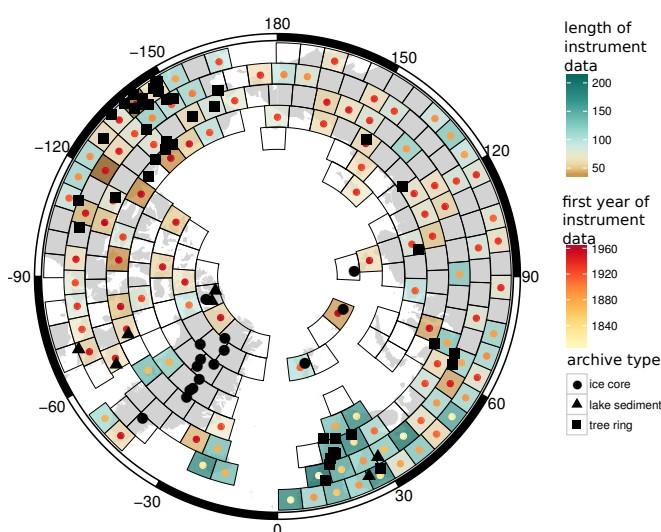

**Figure 1.** Distribution of input data. Length (fill of quadrilaterals) and first year (coloured circles) of the regridded instrumental data. Symbols show the locations and type of used proxy data (PAGES 2k Consortium, 2017). The reconstruction target area are all grid cells marked with wire frames.

## 2.1 Instrumental data

Several different gridded data sets for earth surface air temperatures (SAT) are available from different research groups, derived

from different subsets of instrumental data and presented on different types of grids. Most datasets, like e.g. CRUTEM4 (Jones et al., 2012) or CRU TS3 (Harris et al., 2014) are presented on a regular equilateral grid, such as a $5° \times 5°$ grid. Such a regular grid exhibits severe shortcomings when analysing data close to the poles, as the grid cells become very narrow in meridional direction and almost triangular shaped. One data set, the Berkeley Earth Surface Temperature (BEST) (Rohde et al., 2013), is offered on a $1° \times 1°$ grid as well as an equal area grid. While the latter would be a good fit for the process level model of

BARCAST (c.f. sec. 3), an analysis revealed that this version of the dataset shows rather strange long distance correlations over our region of interest. These might be artefacts of the regridding and interpolation process.

Thus a new gridded instrumental data set is generated for this study. The instrumental data for the CRU TS3.24 (Harris et al., 2014) dataset were downloaded from the CRU website. First, the data were converted into anomalies, using the method of Tingley (2012). The equal area target grid is taken from (Leopardi, 2006). To construct the gridded data, the instrumental data

within each grid cell were averaged, using the variance adjustment scheme described by Frank et al. (2006). In contrast to other methods, no data was shared across grid cells by a prescribed spatial covariance structure or spatial interpolation algorithm. We aimed at retaining the variability that a single instrumental record in the grid cell would exhibit. This is a compromise between an actual grid-cell wide average and the limited spatio-temporal availability of instrumental data in high latitudes. Additionally while some proxies (e.g. the tree-ring maximum latewood density record from Northern Scandinavia by Esper

et al., 2012) represent larger, grid-cell sized regions rather than point estimates (see the discussion in Luterbacher et al., 2016),



none are large-scale regional averages (such as the Central European documentary record of Dobrovolný et al. (2010) used by Luterbacher et al. (2016)).

As can be seen in Figure 1, the resulting instrumental dataset is very sparse in space and time. While ordinary reconstruction methods would indeed struggle with such an input grid, the advantage of the (extended) BARCAST method used here is that

presence and absence of observations is explicitly modelled. The reconstruction target region are land mass containing grid cells only (wire frames in Figure1). This is necessary due to the constraints of the chosen reconstruction method (Tingley and Huybers, 2010a; Werner and Tingley, 2015), more specifically due the homogeneous process level model, which describes the temperature evolution on the grid cell level.

## 2.2    Proxy data

The proxy records (black symbols in Figure1) mostly come from the current version 1.12 of the (PAGES 2k Consortium, 2017) temperature data base, with 6 recently updated ice core records from Greenland with revised and synchronised chronologies, of which three are not in the PAGES2k database (**?**). The data set contains several types of natural archives (tree-rings, ice-cores and marine or terrestrial sediments) and proxy measurements (such as ring width and stable isotopes). Thus the data is sensitive to different seasons, and on different time-scales – partly due to different resolutions and the evaluation procedures, but also

owed to the processes generating the archives. All data north of 60° N contained in the database was selected, with an a priori aim of including all annually resolved records.

As the PAGES 2k Consortium (2017) set out to generate a very inclusive data set, the need arose to again scrutinise the data. A few records were excluded (c.f. table A1), as they did not meet the required response characteristics on actual annual time-scales. Additionally, data was divided into two classes: absolutely and precisely dated tree ring chronologies, and layer-counted

proxies with age uncertainties. The latter comprises varved lacustrine sediments and ice core data. In contrast to the procedure outlined by Luterbacher et al. (2016) tree-ring width measurements are not treated differently from maximum latewood density data, although the spectral properties would in principle warrant this separation (Zhang et al., 2015; Esper et al., 2015; Büntgen et al., 2015).

All of the proxy records used in this study are derived from annually banded archives. While tree-ring records are compiled

by cross-referencing a number of cores for each period, there is usually no replication of ice-cores or varved lake sediments. Thus, these archives can (and usually do) contain age uncertainties (c.f. Sigl et al., 2015) which need to be taken into account. Fortunately, the chosen method (Werner and Tingley, 2015) is able to deal with this issue, provided an ensemble of age models is given for each proxy. Appendix D1 details how these age models are generated. As the majority of the proxy data is more sensitive to summer or growing season temperatures the target season for the reconstruction is summer (here: the months from

June to August, JJA) rather than the annual mean temperature.





## 3   Reconstruction method: BARCAST+AMS

In a recent article, Werner and Tingley (2015) published an extension to the BARCAST method. It extends the work of Tingley and Huybers (2010a), providing a means to treat climate archives with dating uncertainties. The original method has been used in a collection of pseudo-proxy experiments (Tingley and Huybers, 2010b; Werner et al., 2013; Gómez-Navarro et al., 2015),

as well as climate field reconstructions over the Arctic (Tingley and Huybers, 2013), Europe (Luterbacher et al., 2016) and Asia (Zhang et al., 2017).

The method uses a hierarchy of stochastic models to describe the spatio-temporal evolution of the target climate field (here: temperatures) $C_t \in \mathbb{R}^N$ at $N$ different locations throughout time $t$, and the dependence of the observations $O_t \in \mathbb{R}^N$ (proxy data as well as instrumental data) on it:

$$C_{t+1} - \mu = \alpha \left( C_t - \mu \right) + \epsilon_t$$

$$\epsilon_t \sim \mathcal{N}(\mathbf{0}, \boldsymbol{\Sigma}) \quad \text{(independent)} \tag{1a}$$

$$\Sigma_{i,j} = \sigma^2 \exp \left( -\phi |x_i - x_j| \right),$$

The process level is thus AR(1) (1$^{\text{st}}$ order auto-regressive) in time, with an overall mean $\mu$ and the coefficient $\alpha$ modelling the temporal persistence. The year-to-year (or rather summer-to-summer) innovations have an exponentially (with distance between locations $x_i$ and $x_j$) decreasing spatial persistence that is homogeneous in space. The spatial e-folding distance is $1/\phi$. The climate is thus persistent in space and time, and information is shared across these dimensions. This is critical in

constraining age models (see discussion in Werner and Tingley, 2015). The climate process $C$ is never directly observed without error (latent process). The observations are modelled as a noisy linear response function:

$$O_t = \beta_0 + \beta_1 \cdot \mathbf{H}_t \cdot C_t + e_t$$

$$\epsilon_t \sim \mathcal{N}(\mathbf{0}, \tau^2 \cdot \mathbf{I}) \quad \text{(independent)}. \tag{1b}$$

The parameters $(\beta_0, \beta_1, \tau^2, \mathbf{H}_t)$ are assumed to be different for each observation, while in the past one set of parameters was assigned to each proxy type (e. g., tree ring widths, ice layer thickness or isotopic values) (Tingley and Huybers, 2013). The

instrumental observations are assumed to be unbiased and on the correct scale, so that, for this type of observation $\beta_0 = 0$ and $\beta_1 = 1$. The selection matrix $\mathbf{H}_t$ is composed of zeros and ones, and selects out at time step $t$ the locations for which there are proxy observations of a given type. That is, each proxy observation is assumed to be linear in the corresponding local, in time and space, value of the climate.

This data-level model is then refined to include dating uncertainties. To this end, Werner and Tingley (2015) consider the

dependence of the local observations $O_s$ on the local climate:

$$O_s | \mathcal{T}, C_s = \beta_0 + \beta_1 \cdot \mathbf{\Lambda}_s^{\mathcal{T}} \cdot C_s + e_s$$

$$e_s \sim \mathcal{N}(\mathbf{0}, \tau^2 \cdot \mathbf{I}) \quad \text{(independent)}. \tag{1c}$$

The vector $e_s$ is a time series of independent normal errors at location $s$ (c.f. $e_t$ from Eq. (1b)). In analogy to $\mathbf{H}_t$ in Eq. (1b), $\mathbf{\Lambda}_s^{\mathcal{T}}$ is a selection matrix of zeros and ones that picks out the elements of the vector $C_s$ corresponding to elements of $O_s$, and is dependent on the age depth model (ADM) $\mathcal{T}$.



From these model equations, conditional posteriors are calculated. Then, a Metropolis-Coupled Markov-chain Monte Carlo $(MC)^3$ sampler (Altekar et al., 2004; Earl and Deem, 2005; Li et al., 2009) is used to iteratively draw solutions from these posteriors, see (Tingley and Huybers, 2010a; Werner and Tingley, 2015) for details and implementations. In the version implemented here, we modify BARCAST slightly. While Tingley and Huybers (2013) used a single set of response parameters $(\beta_0, \beta_1, \tau^2)$ for all data of one type, and Luterbacher et al. (2016) actually set up a separate observation matrix with a set of

parameters for each single proxy, we choose to update the code. The response parameters are now vectors. While this slows down the computations and also the convergence there is no good reason to assume that all proxies of one type respond in the same way across the whole domain and with know differences in proxy quality.

The reconstruction code is run in 4 chains for 4000 iterations without the age model selection code enabled. By then, the chains have settled to a stable state, and the potential scale reduction factor (Gelman's $\hat{R}$) indicates convergence of the

parameters ($|\hat{R} - 1| < 0.1$). Then, the $(MC)^3$ code of Werner and Tingley (2015) is enabled, and the age models are varied. While this was not necessary in the work of Werner and Tingley (2015), the real world data is much sparser, noisier and does not follow the exact prescribed stochastic model (1a–1c). While this additional step helps speed up convergence it can cause the algorithm to strongly favour one set of age models. This can be checked by analysing the mixing properties over the age models in the heated chains (see discussion in Werner and Tingley, 2015).

## 3.1    Reconstruction Quality

The reconstruction calibration and validation statistics are shown in appendix A. Both the $\overline{\text{CRPS}}_{\text{pot}}$ (which is akin to the Mean Absolute Error of a deterministic forecast, see Gneiting and Raftery (2007)) as well as the Reliability score (the validity of the uncertainty bands, Hersbach, 2000) show a decent reconstruction quality (Figure A1 top row). The $\overline{\text{CRPS}}_{\text{pot}}$ on average shows a mismatch of $0.1°\text{C}(0.3°\text{C})$ in the calibration (validation) interval and the Reliability is mostly better than $0.1°\text{C}$. Additionally,

a probabilistic ensemble based version of the coefficient of efficiency (CE) and the reduction of error (RE), (Cook et al., 1994) are generated. These show a skillful reconstruction in most grid cells containing instrumental temperature data – at least in regions where proxy and instrumental data are present over most of the validation period. Note that the quality of the instrumental data, or rather the representativeness of (often) a single meteorological station record can be debated. In fact, in contrast to other BARCAST based reconstructions the one presented here shows a substantial ($\tau_I^2 \approx 0.15$ in standardised units)

noise level for the instrumental data. As other gridded instrumental datasets employ spatial interpolation processes they are generally smoother in space than the gridded instrumental dataset generated for this study. Thus these gridded products are closer to the spatial part of the process model in Eq. (1a).

Another means of assessing the reconstruction quality is to check the variability or spread of the different ensemble members in space and time (see appendix B). The effect of the spatially and temporally sparse data can easily be seen in Figures A2 and

A3, clearly indicating the increased uncertainties back in time and in space in the absence of proxy data. This analysis hints that while there could still be skill left in the mean Arctic summer temperature reconstruction in the first centuries CE, the precision of the spatial reconstruction rapidly decreases in areas that become more data sparse. While the reconstruction over the regions with local proxy data present – such as Fennoscandia – remains reliable, a time-varying reconstruction domain (or rather,



domain over which the reconstruction is analysed) would unnecessarily complicate things. Thus the gridded reconstruction is
only shown back to 750 CE. However, for single analyses over data rich regions the full reconstruction period (1–2002 CE) can
be used.

Additionally, the spectral properties of both the reconstruction and the proxy input data are analysed (c.f. appendix C). Not
all proxies contain signal on centennial or longer time scales, and the reconstruction method explicitly describes year-to-year
summer temperatures as an AR(1) process. Thus, the reconstruction shows properties of an AR(1) process over most of the
reconstruction domain. However, when comparing the area averaged temperature reconstructions against other multiproxy
reconstruction data one can still see similar variability on centennial and longer time-scales (see Figure 3).

## 4 Results

### 4.1 Mean Arctic Results

The area averaged summer temperature reconstruction is shown in the bottom half of Figure 2 as the point-wise (annual
summer) ensemble mean (heavy blue line). The first millennium CE shows a mean reconstruction that is relatively flat, with
an apparent change in variability. This is caused by the increased variability between the different ensemble members and
thus by the reduction in proxy data coverage back in time. The effect of the spatial proxy data coverage on the reconstruction
intra-ensemble variance is further discussed in Appendix B.

The new spatially averaged SAT reconstruction shows a pronounced variability on a broad range of time-scales. The longer-
term, centennial to millennial, evolution of the reconstructed SAT demonstrates a reasonably good agreement with a general
pattern that was inferred in previous temperature reconstructions for the Arctic and its sub-regions (Figure 3). Throughout
most of the reconstruction period, the Arctic SAT anomaly shows an overall orbital forcing-driven cooling trend. We note
that the cooling trend reversal occurs already in the first half of the 19[th] century as a recovery after the last LIA minimum.
The attribution of the actual timing depends on the considered time-scale and the use of 1850 CE as an upper limit for the
trend magnitude estimate is therefore somewhat arbitrary but allows comparison with the previous studies on the topic. The
observed trend over the period 1-1850 CE is, however, not statistically significant and its magnitude of -0.01±0.01 °C/century
for the pan-Arctic average over the terrestrial domain is substantially lower than the values within the range of -0.03 to -0.05
°C/century inferred from the previous Arctic2K reconstructions (PAGES 2k Consortium, 2013; McKay and Kaufmann, 2014)
(see summary in Figure 4b). Moreover, Figure 4 demonstrates that the trend is spatially inhomogeneous. The largest magnitude
of the millennial-scale cooling of $-0.5 \pm 0.1°C$ over the period 1-1850 CE is registered in the comparatively proxy data-rich
regions between 0–100° E and 110–170° W. The Greenland region and the Canadian Arctic, in contrast, show a warming with
a mean temperature rise of $+0.5 \pm 0.1°C$ before 1850 CE. Meridional trends in the latitude-averaged reconstruction displayed
in Figure 4b reveal a similar pattern for the region within 110–180° W with a preferentially cooling trend, yet more variable
tendencies to cooling/warming are apparent for the rest of the circum-Arctic domain. We note that except for the meridional
sector of 40–45° W (Greenland ), none of the zonally averaged reconstructions show a statistically significant change for the
considered period 1-1850 CE.





**Figure 2.** a) Ensemble-based mean Arctic summer (June–August) SAT (land only) anomaly probability density for the three selected centennial-scale and three decadal-scale time periods. For comparison the mean Arctic SAT anomaly for the entire reconstruction period is also presented. Note that the mean for the last decade (2006–2015 CE) is based on the infilled instrumental data only. b) Arctic (land only) average summer temperature anomalies over the instrumental period and number of instrumental observations available. Grey box denotes the calibration interval. c) Ensemble-based spatially averaged time variability of the seasonal SAT probability distribution over the reconstruction period, blue line: ensemble mean, shading: (pointwise) 95% posterior. Red line: instrumental data. Note that before 1870 CE the number of instrumental observations rapidly decreases. d) Number of proxies by archive type over time.





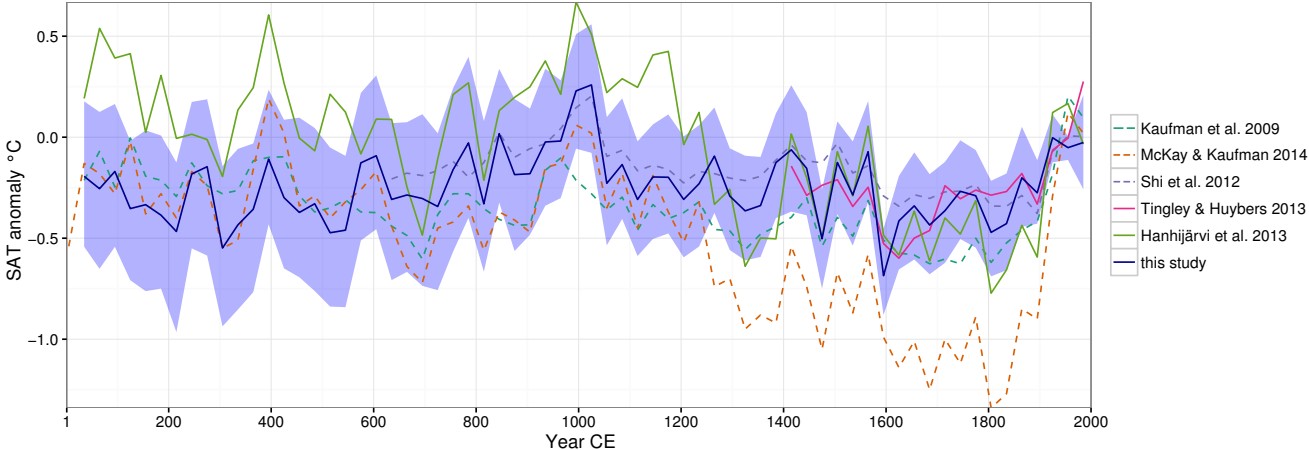

**Figure 3.** Comparison of this reconstruction (with 95% confidence band) with other reconstructions. 30 year averages. Note that McKay and Kaufmann (2014) target annual temperatures and Hanhijärvi et al. (2013) reconstruct the North Atlantic sector of the Arctic ($50^\circ$ W $- 30^\circ$ E).

The proxy dataset from Greenland is dominated by oxygen isotopes series from ice cores. These are not entirely coherent in the magnitude and sign of the millennial trend, and in some cases reflect more annual rather than summer season SAT. As a result the detected warming could stem from the different weights of the proxy records in the reconstruction procedure. The

oxygen isotope series are also known to be subject to a possible warm bias due to increased storm activity which might accompany the LIA (Fischer et al., 1998) and/or be influenced by the site and source temperature compensating effects (Hoffmann et al., 2001; Masson-Delmotte et al., 2005) that could actually mask the LIA cooling. Meanwhile, the north of Greenland is less influenced by precipitation associated with North Atlantic storms, though lasting seasonal accumulation changes could be another source of uncertainties important especially on longer time-scales: ice cores from northern Greenland are expected

to have a higher fraction of summer precipitation than those from the south due to the effect of continentality on the annual accumulation, and hence exhibit a higher sensitivity to summer conditions. The overall LIA cooling might have amplified this effect, thus promoting a higher percentage of istotopically warmer summer moisture in the annual accumulation, this way diminishing or even reversing the cooling trend. While site and source temperature compensating effects for the individual series can be accounted for by using the records of deuterium excess (Masson-Delmotte et al., 2005), other potential biases are

difficult to resolve without additional support e.g. from general circulation models.

     Superimposed on the trend are the two major centennial to multi-centennial scale anomalies: the Medieval Climate Anomaly – a warm period during the 10[th] and 11[th] centuries CE, and the two phases of the cold Little Ice Age between ca. 1100–1500 CE and 1550–1900 CE. Our reconstruction suggests the coldest centennial-scale period of the LIA was during the 15[th] and 17[th] centuries CE. The coldest decadal-scale event in the reconstruction, however, occurred in the early 19[th] during the Dalton

Minimum of solar activity and a period of increased volcanic activity with two major tropical eruptions of 1808/1809 CE and Tambora 1815. The slow millennial-scale cooling is finally terminated by an abrupt contemporary warming which is clearly identifiable since the beginning of the 20[th] century.





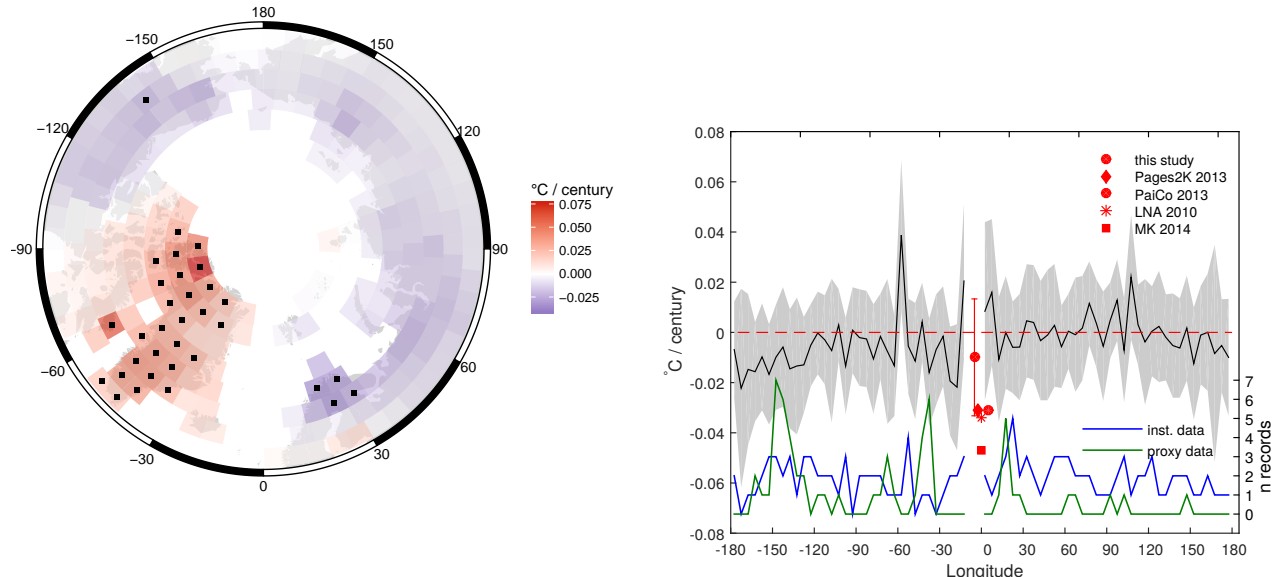

**Figure 4.** (a) Ensemble-averaged spatial linear trends over the period 1–1850 CE in °C/century; black dots mark locations where the trend is statistically significant for more than 95% ensemble members; (b) Ensemble-averaged meridional trends in the latitude-averaged reconstruction over the period 1–1850 CE (solid black line); meridional averaging over the 5° segments and zonal over the terrestrial nodes is applied to each reconstruction ensemble member. Grey shading highlights the ±2*std interval on the estimated trend magnitude derived from the ensemble of spatially averaged reconstructions. Solid green and blue lines show the number of proxy and instrumental records in each 5° longitudinal sector. For comparison, the pan-Arctic trend estimates for the same period are shown: Pages2K 2013, PaiCo 2013 and LNA 2010 PAGES 2k Consortium (2013), MK 2014 McKay and Kaufmann (2014).

Figure 3 suggests that the LIA cooling is less pronounced in the new reconstruction compared with the same period in McKay and Kaufmann (2014). A likely explanation of this difference is the effect of targeting the summer season (as in our study) compared to annual mean in the reconstruction of McKay and Kaufmann (2014). Throughout the LIA sea-ice cover has most likely experienced a pan-Arctic expansion as evidenced proxy studies (e.g. Belt et al., 2010; Kinnard et al., 2011; Berben et al., 2014; Miettinen et al., 2015) and also supported by documentary evidence for the last phase of the LIA (Divine and Dick, 2006; Walsh et al., 2017). Such sea ice expansion would lead to an increased continentality of the climate in most of the study domain, implying larger summer to winter SAT contrasts (see e.g. Grinsted et al., 2006, for Svalbard), with the correspondent effects on differently targeted reconstructions and the inferred magnitude of LIA cooling.

Comparing the magnitude and spatial extent of past warm periods featuring similar settings in external forcing with the present-day warming is of major importance, since it provides possible limits for the scales of naturally forced climate fluctuations. Figures 2 and 3 suggest that in the new reconstruction, the period around 1000 CE, typically associated with the peak of the MCA, show up at least similarly warm as the reconstruction for the late 20[th] and early 21[st] century, although the



instrumental data suggest much warmer temperatures in the last decade (2006–2015 CE). This is in accordance with the con-
clusions reached previously in Shi et al. (2012), Hanhijärvi et al. (2013), and McKay and Kaufmann (2014). At the same time,
in contrast to other reconstructions (notably Hanhijärvi et al., 2013) the Roman times around the first and second century CE
do not show up as particularly warm in the circum-Arctic mean, which is also reflected in the analyses presented in section 4.2.
Note that their reconstruction was actually limited to the North Atlantic sector of the Arctic. Additionally the spatial skill of
the reconstruction decreases back in time as the proxy data becomes sparser (see appendix B), and spatial averages thus result
in higher uncertainties and will be closer to the overall mean.

Figure 2a presents the ensemble-based probability densities (pdf) of the spatially averaged across the reconstruction domain
mean SAT anomalies for the six selected reconstruction sub-periods. These are the three selected century-long periods of 960–
1060 CE, 1580–1680 CE and 1903–2002 CE, representing the two warmest and coldest century-long periods of the record and
the last century-long period of the reconstruction that includes Contemporary Warm Period (CWP, in this study since 1978 CE
onwards). For comparison the same pdf for the entire reconstruction period is also presented. To further highlight the contrasts
between the mean and extreme climate states, pdfs for the four shorter decadal-scale intervals corresponding to anomalously
warm and cold periods (see Subsection 4.2 for details) are displayed, including the most recent warm decade of 2006–2015 CE.
For the latter we used an instrumental grid infilled by BARCAST. The maps of spatial mean SAT anomalies for these periods
follow in Figure 5.

The coldest phase of the LIA with a mean centennial-scale SAT anomaly of $-0.5 \pm 0.1°$C vs. MCA $0.1 \pm 0.1°$C and CWP
$0.03 \pm 0.05°$C emphasises a difference between the extreme warm and cold century-long periods in terms of the pan-Arctic
summer temperature probability density. Figure 2a) suggests that the centennial-scale maximum of the MCA could be at least as
warm as the CWP, although a reduction of proxy data after the 1990s likely introduce a cold bias when estimating present-day
warming in the reconstruction.

In order to quantitatively test the significance of the observed reconstructed differences in SAT anomalies between the
selected periods, the two-sample $t$-test is used on the samples of the derived distributions. During the testing procedure the
realisations from different ensemble members of the Arctic SAT annual means are not pooled. Rather, the respective pdfs for
the selected periods are derived for every individual ensemble member of the reconstructed SAT. The procedure uses bootstrap
estimates of the pdf for the period (MCA and CWP) averages derived from 100 independent draws. The two-sample t-test
with separate variances is applied to test the null-hypothesis of the two samples associated with the two different warm periods
to originate from two normal distributions with equal means and unknown and non-equal variances. Using a one-tailed $t$-test
should then provide information on whether the MCA was on average warmer or colder than the last 100 years. The test
statistics for each ensemble member is then collected and analysed.

The testing results for a two-tailed test rejected $H_0$ of equal Arctic mean SAT anomalies between 960–1060 CE and 1903–
2002 CE for 98% ensemble members. No conclusive answer can however be reached for the sign of the bias in the means
between the centennial periods. Although the MCA appears warmer on a centennial time-scale compared with the last 100
years as shown in 2a), testing fails to reject the null-hypothesis for 7% of the ensemble members, whereas for the opposite
alternative hypothesis (i.e. CWP warmer than MCA on average) the hypothesis rejection rate is as high as 6%. Taking the



multiple testing into account, we conclude therefore that given the collection of the proxy and instrumental data, and the
reconstruction technique used, it is not possible to infer whether the Arctic summers of last 100 years of the reconstruction (i.e.
before 2002 CE) were unprecedentedly warm when compared with the previous major warm climate anomaly. We note also
that higher variability in the derived ensemble of realisations for the mean Arctic SAT anomaly during the warmest intervals
of the MCA of $0.5 \pm 0.2°$C and CWP of $0.2 \pm 0.3°$C similarly prevents from reaching any firm inference on the relative

magnitudes of the two decade-long anomalously warm periods of the new reconstruction.

The scale of the warming north of $60°$that occurred over the last decade (2006–2015 CE) is hinted at by the pdf of the (in-
filled, gridded) temperature data. It suggests a positive SAT anomaly of $1.0 \pm 0.05°$C, which appears significantly warmer than
the reconstructed decadal temperature maximum of the MCA. However, first of all the comparison of the two independently
infilled processes is not advisable. The underlying stationarity assumptions about systematic biases from the infilling are vio-

lated in this case, and additionally there is a tendency to underestimate extremes in the past due to sparse data, but also due to
the assumed AR(1) process model. Thus, while hinting that the last decade could have been unprecedented over the last 1100
years, the exact amplitude of this current warming in the instrumental data with respect to that in the maximum of the MCA
remains more uncertain than the pdfs would suggest at face value.

### 4.2 Spatial signature of past and recent extreme temperature anomalies

The distribution of extremely warm and cold years in both space and time is analysed by ranking the years according to
their seasonal temperature for each ensemble member and the reconstruction node. Due to insufficient proxy data density and
hence the inflated intra-ensemble variance (see Figure A2) in the early part of the reconstruction period, the analysis is limited
to the time after $750$ CE. The probability density for each year to be ranked as warmest or coldest is calculated across the
entire ensemble for each individual location. Further marginalisation over the spatial domain yields a probability density of a

particular year to be associated with an Arctic-wide warm or cold extreme. In order to check the statistical significance of the
derived probability densities, the analysis is replicated on an ensemble of surrogates derived from the original reconstruction
ensemble using bootstrapping along the time axis. The surrogates were constructed in a way to mimic the statistical properties
of the original reconstruction, such as a local serial correlation, variance and a spatial coherence. The derived time-average
$[0.025\ 0.975]$ percentiles of the spatial probability densities for the bootstrap surrogates are then used as the respective quantiles

for marking the years as potentially coldest or warmest during the analysis period (Figure 6). For convenience, the probability
densities are further marginalised over the 5 degree longitude bins and presented as a time–longitude colour map in Figure
7a. Additionally, potentially warmest and coldest decades using the spatially marginalised probability density functions are
evaluated (Figure 7b). The significance testing procedure for the decadal extremes is based on a similar analysis.

The results of the analysis are reflective of the longer term (millennial and secular) pan-Arctic tendencies in the seasonal SAT,

yet the inter-regional differences are made clear as well. Of the series of past and present exceptional warmings, compared with
the part of the present-day warm period before 2002 CE, the peak of the MCA features the two phases of the same pronounced
pan-Arctic warming with a consecutive series of spatially coherent warm extremes between ca. 935–1035 CE (Figure 7a). On
decadal time-scale (Figure 7b) the MCA is marked over the whole region by anomalies having a persistent high fraction of



**Figure 5.** Ensemble average of the reconstructed Arctic2k SAT anomalies over the century-long periods of (a) 960–1060 CE, (b) 1580–1680 CE and (c) 1903–2002 CE; (d) Ensemble average SAT anomaly over the period of 982–991 CE, a potentially warmest decade since 750 CE with nine of ten years top ranked as potentially warmest; (e) Ensemble average SAT over the period of 1812–1821 CE within six out of ten years are ranked potentially coldest; (f) Ensemble average SAT anomaly over the period of 1993–2002 CE with 12 out of 16 years ranked as potentially warmest. Colours show the temperature anomalies. Proxies marked by black dots.





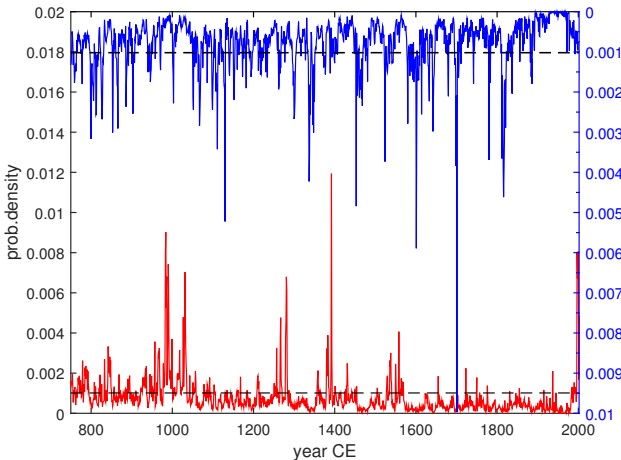

**Figure 6.** Probability density of statistically significant, potentially warmest (red) and coldest (blue) years over the 2000-year long period marginalised over the reconstruction domain. Dashed lines indicate the 95% significance level estimated from the ensemble of bootstrap surrogates.

likely warmest decades with a negligibly low fraction of coldest decades. In particular, nineteen of twenty years during the two
decades of 981–1000 CE and all ten years of the decade 1026–1035 CE were ranked as statistically significant warm extremes among the ensemble members, which makes these two periods potentially warmest in the reconstruction since at least 750 CE.

Figure 7a also highlights a difference in time evolution of the regional expression of the MCA via spatial incoherence of extremely warm years/decades during the period associated with this climate anomaly. A somewhat earlier onset of warming in the Asian domain is evident from an increased frequency of warm extremes between 45° E–160° E, followed by a coherent
warming in the Greenland and North Atlantic (NA) sector of the study domain. Figure 5d exemplifies a picture of a pan-Arctic warming during the first warm decade 982–991 CE of the first phase of the MCA with the largest reconstructed positive anomalies attained within the 80–130° E domain.

Figure 7b demonstrates that the period after the MCA termination features a variable climate as manifested by an alternating sequence of potentially warmest and coldest decades lasting until approximately 1560 CE. Despite that this period is generally
associated with the colder LIA, remarkable warm decadal-scale periods are detected between 1273-1289 CE, 1375-1393 CE and 1528-1540 CE. The following transition into the coldest phase of the LIA after 1560 CE is clearly marked by a drop in the frequency of potentially warmest decades to zero, whereas the cold decades dominate until the onset of the contemporary warming after ca. 1980. Figure 5e shows the spatial ensemble average SAT anomaly for the potentially coldest decade of the LIA of 1812–1821 CE, with all ten years ranked potentially coldest across the entire reconstruction ensemble. Note that
Greenland and Baffin Island and the surrounding areas during this decade remain relatively warm in contrast to the rest of the reconstruction domain. This local anomaly agrees well with the abrupt minimum of spring sea-ice extent on South-East Greenland shelf during the early 19[th] century, reconstructed from the high-resolution sediment core MD99-2322 Miettinen





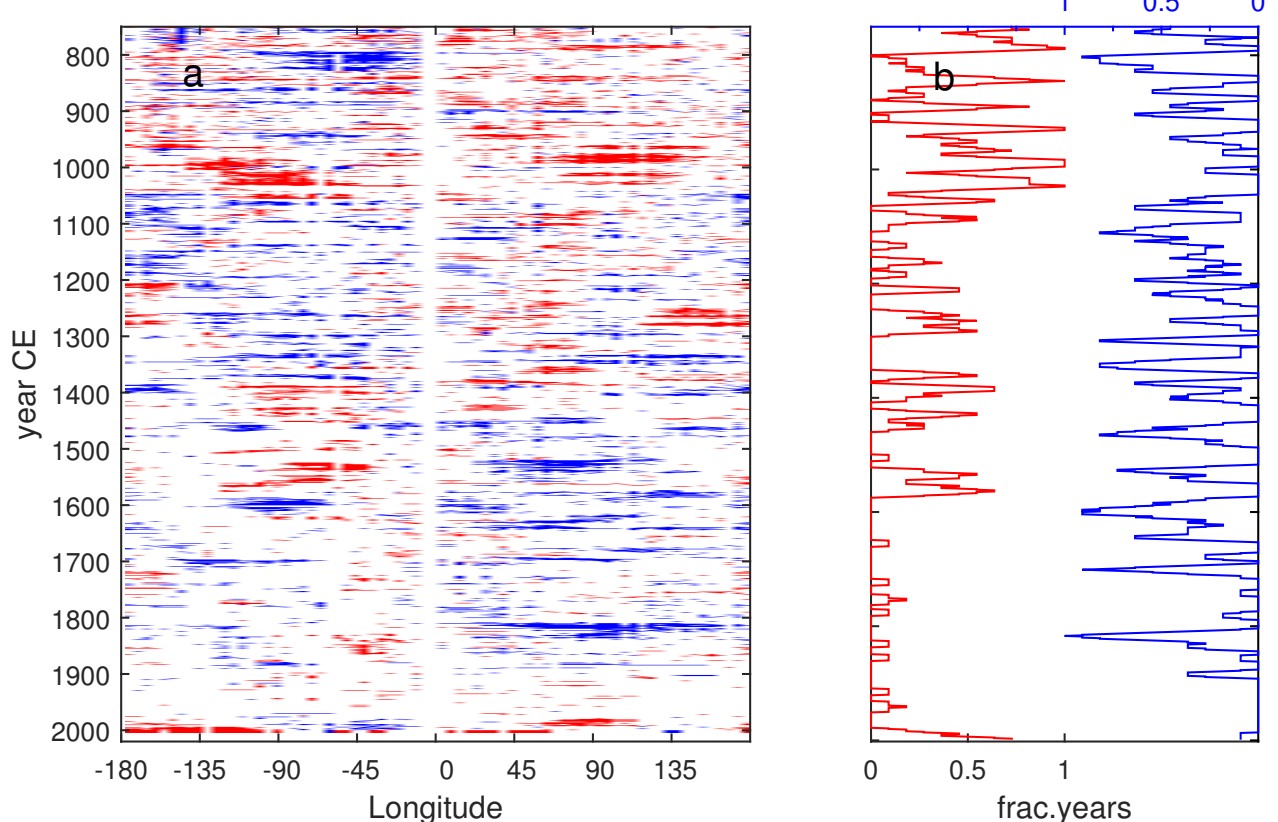

**Figure 7.** Left: Occurrence of statistically significant, potentially warmest (red) and coldest (blue) years over the 2000-year long period in different sectors of the Arctic domain. Right: Fraction of potentially warmest/coldest decades with respect to time. Both MCA and LIA clearly shows up.

et al. (2015). We suggest this can be attributed to the effect of a strong negative North Atlantic Oscillation (NAO) anomaly during this period, possible in combination with a change in the Subpolar gyre circulation (e.g. Schleussner et al., 2015; Otterå et al., 2010) triggered by a sequence of volcanic eruption in the early 1800s including the unknown 1809 event and the major Tambora 1815 eruption.

Contemporary warming is manifested as a sequence of potentially warmest years starting in 1990 CE within 45–100° E and since 1995 propagating to almost the entire reconstruction domain between the longitudes of 180° W to 100° E. Figure 5f shows a spatial map of temperature anomalies for the period 1993–2002 CE, that features 6 out 10 statistically significant warm extremes on the pan-Arctic scale. When compared with the probability density marginalised over the spatial domain displayed in Figure 6, contemporary warming clearly reveals a similar coherence both in the spatial domain and agreement over the range of ensemble members, compared with the estimates made for the MCA. One should emphasis that this statistically significant sequence of warm extremes was detected outside the calibration period, which provides another indirect proof for a skill of our





new Arctic2k reconstruction. This reconstruction, however, does not extend into the very last 15 years, over which warming in
teh Arctic has been continuing (c.f. Figure 2). With these years included in the analysis, the signature of the CWP would much
likely become more prominent (see discussion in Section 4.1).

### 4.3 Arctic amplification in the European sector of the Arctic during the MCA and contemporary warming

The concept of "polar amplification" (of which Arctic amplification is the Northern Hemisphere expression) predicts a larger
magnitude of warming in the polar regions during lasting warm periods than the one observed on a global or hemispheric
scale (Hind et al., 2016). As the proxy data is sparse in space and time, we here focus on the regional expression of the
Arctic amplification by analysing the previous warm anomaly, the MCA, over the European sector. While there are gridded
temperature reconstructions over North America (Wahl and Smerdon, 2012) and Asia (Zhang et al., 2017), the North American
one goes back only to 1200 CE, and there is only a low number of proxy records in the Asian sector of the Arctic.

The analysis is based on the EuroMed2k reconstruction, the ensemble–based gridded summer temperature reconstruction
by Luterbacher et al. (2016), who use a similar reconstruction method. While both gridded reconstructions (the one presented
here and that by Luterbacher et al., 2016) are skilful on the grid-cell level only back to around 750, averages over latitudinal
bands will retain skill prior to that. The reconstructions are averaged over the three latitudinal bands 30–45° N, 45–60° N and
60–70° N within the longitudinal sector of 0–50° E. Then, the ensembles of centennial anomalies both for specific periods
and in a sliding window in the time domain are calculated, using a coldest period in the Arctic2k reconstruction of 1580–
1680 CE (Figure 5b) as a reference for deriving the respective regional centennial changes. In order to test the hypothesis of the
two samples to have statistically significant means a paired-sample $t$-test was used, implicitly assuming the samples are nearly
Gaussian distributed. Note that a direct comparison of the Arctic2k results with the EuroMed2k reconstructions are problematic
due to different calibration periods used in BARCAST for these reconstructions; the inference is therefore mainly based on
the European reconstruction alone. Results for the Arctic2k reconstruction, however, provide an outlook to the intra-ensemble
spread on centennial time-scale and the time evolution in the respective centennial estimates relative to the reference period.

Figure 8 shows the spatially averaged probability densities of century mean SAT anomalies for the three centuries overlap-
ping with the MCA. For comparison, similar results are also presented for the contemporary warming using the last possible
centennial period of 1903–2002 CE. Although in the European sector the effect of the Arctic amplification is evident for all
four century-long periods, the 11[th] century clearly stands out, featuring a seasonal SAT anomaly relative to the defined baseline
in the 17[th] century of $0.8 \pm 0.1°$C at 60–70° N. For comparison, the observed centennial warming at 30–45° N had a magnitude
of $0.2 \pm 0.1°$C, yielding a $0.6 \pm 0.2°$C statistically significant warming amplification across the considered range of latitudes.

While the Arctic amplification is apparent already at the peak of the MCA warming (anomaly of $1°$C at 60–70° N vs.
$0.7°$C at 30–45° N), the analysis of the time evolution of centennial mean SAT anomaly shown in Figure 9 demonstrates the
rather dynamical behaviour of Common Era temperature variability. This underlines the necessity to look beyond the usual
analysis of equilibrium states (c.f. Hind et al., 2016). While the latitudinally banded temperature maximum was reached almost
concurrently, with a hard to quantify lag at high latitudes, the cooling at higher latitudes lags behind the lower latitudes.
Thus, there is an apparent warm amplification, which should rather be attributed to a lagged and slower cooling at higher




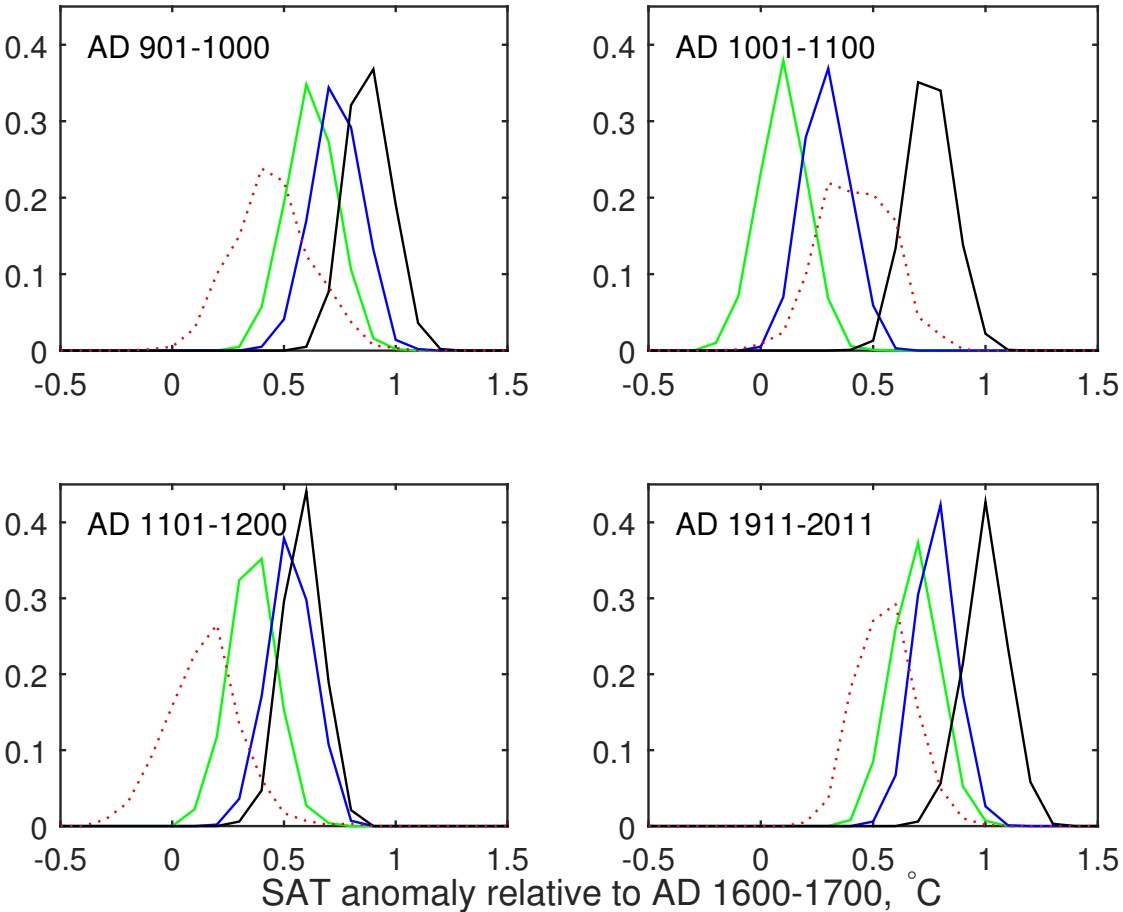

**Figure 8.** Centennial-scale signature of Arctic amplification during the MCA and contemporary warming demonstrated via comparison of temperature differences between the cold 17th century and warmer periods of 10th–12th and 20th centuries across the latitudes. The probability densities are calculated as ensembles of centennial mean anomalies over the four latitudinal bands of 30–45° N (green), 45–60° N (blue), 60–70° N (black) and >60° N (red). For the first three bands averaged over 0–50° E lower latitude PAGES2k reconstruction over Europe (Luterbacher et al., 2016). Note that our Arctic2k reconstruction uses different calibration interval what hinders a direct comparison of the respective SAT anomalies distributions derived from other reconstructions.





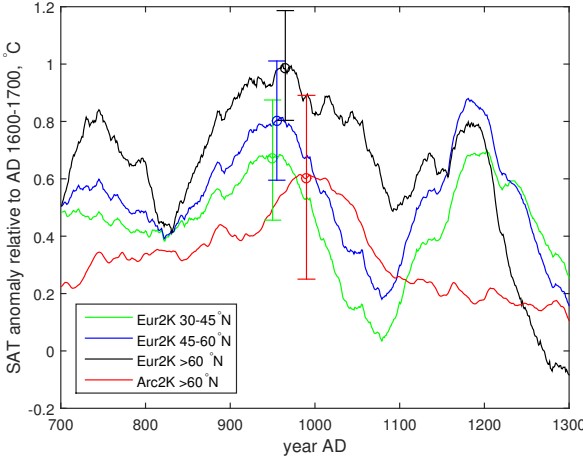

**Figure 9.** Time evolution of ensemble averaged centennial mean SAT anomaly over the period of 700–1300 CE for the four latitudinal bands in the European sector 0–50° E of the Northern hemisphere. Latitude average SAT anomaly for the three bands south of 60° N were calculated using the PAGES2k reconstructions over Europe (Luterbacher et al., 2016), while our Arctic2k reconstruction is used for the Arctic higher latitudes (>60° N). Error bars highlight the ensemble based CIs for the timings corresponding to the warmest year of the MCA at the respective range of latitudes. The anomalies were calculated relative to the respective centennial means for the coldest centennial-long period of the LIA, 1580–1680 CE.

latitudes of the European Arctic. This cooling commenced after the centennial average temperature maximum of the MCA (950–1000 CE). The multi-decadal time-scale of the observed lag points to a particular role of oceanic mechanisms in creating

and maintaining the positive temperature anomaly after the LIA temperature maximum. Notably the cooling following the next less pronounced MCA-associated maximum around 1200 CE demonstrates the opposite pattern with the temperatures at the higher north showing a more rapid decline during the 13th than at the lower latitudes, although the magnitudes of warming at higher $0.6 \pm 0.1$°C vs lower $0.4 \pm 0.1$°C latitudes remain statistically different.

Figure 8d shows that the modern period features a statistically significant offset of $0.4 \pm 0.2$°C between the higher ($1.1 \pm$

$0.1$°C) and lower ($0.7 \pm 0.1$°C) latitudes in the European sector, which is still less than the one registered in the reconstruction during the MCA maximum. The Arctic2k reconstruction however shows a more pronounced warm bias specifically for the last century (see the respective probability densities on panels (b) and (d) in Figure 8). Whether this is an artifact of time variability in the reconstruction quality or an indication/confirmation of an unprecedented decline in sea-ice extent during the recent decades still needs to be clarified.

**5 Discussion and conclusions**

This paper presented a new circum Arctic CF reconstruction of summer season temperatures back to 750 CE with the Arctic average SAT anomaly extended back to 1 CE. The reconstruction uses a subset of 54 annually dated temperature sensitive



terrestrial proxy archives of various types mainly from an updated and corrected PAGES2k database supplemented with six recently reanalysed (**?**) Greenland ice core series.

The technique applied is a recent extension of the Bayesian BARCAST which provides a means to explicitly treat climate archives with dating uncertainties which previously would be used on their "best guess" chronologies. Another added value of using the Bayesian technique was a generation of the ensemble of 1400 equally likely, independent realisations of past CF evolution that enabled us considering the past climate regional variability in a probabilistic framework. Therefore this new Arctic2k reconstruction is essentially an ensemble of possible realisations of the past Arctic summer climate given the data and
the reconstruction technique used.

    The quality of the reconstruction in the spatial and temporal domains was tested using a suite of metrics such as continuously ranked probability score ($\overline{\text{CRPS}}_{\text{pot}}$) and the reliability (Reli) score which are more appropriate for the Bayesian framework than the "Coefficient of Efficiency" and "Reduction of Error", which are typically used in palaeoclimate research. Judging from these scores it could be demonstrated that the new reconstruction is skillful for the majority of the terrestrial nodes in
the reconstruction domain making it a useful product for studying the late Holocene Arctic climate variability at the region scale. However, from the analysis of intra-ensemble variability, but also from analyses on the extreme years and the calculated confidence intervals the reduction of skill back in time is apparent. This is mostly caused by the proxy network, which is getting sparser when going back in time, and should be taken into account when the new reconstruction is used for making any quantitative inferences.

Although this study is mainly focused on presenting the new reconstruction and assessing its quality, we also ran some basic quantitative data analysis in order to uncover the potential of the new product and consider the results in light of previous studies on the subject. In particular, the area averaged Arctic2k reconstruction features similar major cold and warm periods throughout the last two millennia and thus compares favourably with earlier studies targeting a similar season and region.

    The major findings from the analysis of the new reconstruction are as follows:

There is a notable lack of pronounced orbital scale cooling trend over the Common Era – a period over which the summer insolation has mostly been decreasing. In agreement with findings of Nicolle et al. (2017), our results demonstrate this orbital cooling trend is not necessarily observed in all sub-regions, although Nicolle et al. (2017) combine the data from around the North Atlantic realm into a single composite series. The magnitudes for the local cooling does not exceed $-0.5 \pm 0.1°$C over the period 1-1850 CE and this change is found statistically significant only in the comparatively proxy data-rich regions between
$0$–$100°$ E and $110$–$170°$ W. In contrast to this, the reconstruction shows that the Greenland region and the Canadian Arctic even exhibit a warming with a mean temperature rise of $+0.5 \pm 0.1°$C before 1850 CE. While this can in part be explained by controls on the stable water isotopes in the annual accumulation other than mean ambient temperature, there is also the possibility for contrasting decadal to century long anomalies in oceanic and atmospheric temperatures between the eastern and western subpolar North Atlantic due to variations in the relative strength of the eastern and western branches of the AMOC (e.g.
Schulz et al., 2007; Hofer et al., 2011). In particular, Seidenkrantz et al. (2007) report an oceanic warming south of Greenland during the northeastern North Atlantic cooling episodes such as the LIA, indicating an increased advection of warm Atlantic water by the West Greenland current. Miettinen et al. (2012) and Thornalley et al. (2009) show continuous warming trends in





the central subpolar North Atlantic over the late Holocene driven by lasting oceanic circulation changes. Though the evidence of persistent changes in their strength during late Holocene are still rather inconsistent, this can still be considered as a possible
contributing mechanism behind the observed lack of millennial cooling in the reconstruction for the Greenland region.

      The analysis of the reconstruction reveals the spatial signatures of the two major climate anomalies of the last two millennia, the LIA and MCA, as well as the beginning of the CWP in the circum-Arctic region. Although there is evidence for prominent and lasting climate anomalies in the pre-750 CE period too, these results should be interpreted cautiously due to the drastic reduction in proxy data density in the early part of the reconstruction period. The MCA in the circum-Arctic region can be
associated with a century-long period between ca. 960–1060 CE showing an area average SAT anomaly of $0.1 \pm 0.1°$C. The MCA features two temperature maxima and different spatial extent of the regional anomalies that progress from the Eurasian to the North American segments of the Arctic realm. A coherent warming of the period 982–991 CE yields the warmest decade of the MCA with the area average summer temperature anomaly of $0.5 \pm 0.2°$C. While the most recent warming shows an even stronger regional coherence than the MCA, even across continents (PAGES 2k Consortium, 2013), the MCA was still an
unusual and extremely warm period in the context of the past two millennia. However, given the input data available and the reconstruction method used it cannot be decided with any statistical significance whether the MCA or the CWP was warmest in the reconstruction.

      The new reconstruction suggests a relatively long, interrupted by abrupt decadal-scale warmings, transition to the LIA after the second of the two MCA maxima ends at around 1060 CE. The coldest century-long period of 1580–1680 CE shows an
almost spatially coherent circum-Arctic summer cooling with a remarkable exception in north Greenland. While the coldest century in the LIA (and in the reconstruction as a whole) was in the 17[th] century CE, the cooling went on until the mid 19[th] century CE. Most of the Arctic was coldest during the decade of 1812-1821 CE following the 1809 (unknown) and 1815 (Tambora) eruptions, which caused the "Year without a Summer" in 1816 over most of Europe and yielding a circum-Arctic SAT anomaly of $-0.8 \pm 0.2°$C. The cooling was, however, not pan-Arctic: warmer than normal conditions were reconstructed
for Greenland. From independent proxy data it is known that during that time, the sea ice extent could be lower than average (Miettinen et al., 2015) which, again, could be a signature of the ocean circulation response triggered by negative anomalies in the radiative forcing (Schleussner et al., 2015; Otterå et al., 2010).

      The comparison of the time-variable behaviour of summer temperature anomalies between high latitudes and lower latitudes shows the very dynamic characteristics of the Arctic amplification. In addition to the amplified temperature anomaly
in the high north compared with lower latitudes, during MCA this phenomenon also shows up as a clear multi-decadal lag in summer temperature maxima in the highest latitudes when comparing against mid latitude European summer temperature reconstructions.

      The lack of a pronounced orbital cooling trend throughout the Common Era, and the spectral characteristics of both proxies and reconstruction show that still work is needed on generating more and longer high quality proxy series in parallel with
a reanalysis of the existing data. Especially updating the mostly only half a century long North American tree ring series towards the present (Babst et al., 2017), but also possibly extending some of them into the first millennium CE seem to us like worthwhile efforts. Moreover, a relative "flatness" of spectra on sub-decadal to multi-decadal time-scales contrasting with an



inflated variance of the multi-decadal to millennial variability (Appendix C) for some of the tree-ring chronologies, suggests that a reassessment and potentially a revision of the raw data processing techniques used for this chronologies would be highly

desirable.

BARCAST as a CF reconstruction technique still offers a large potential for future development and use in new improved reconstructions. In addition to including explicitly the annually dated proxies with the chronological uncertainties into the scheme, what became a major innovation of the presented reconstruction, the next natural step will be a development of a theoretical and numerical framework to extend the technique to non-annually dated proxy archives with chronological uncertainties.

This will enable a substantial extension in the proxy coverage both in the spatial and time domains including the marine realm dominated by non-annually resolved marine sediment proxy archives, potentially promoting an improved performance of the reconstructions at the low-frequency (centennial) time-scales.

*Acknowledgements.* J.P.W. gratefully acknowledges support from the Centre for Climate Dynamics (SKD) at the Bjerknes Centre. D.V.D. contribution to the Arctic2k was partly supported by Tromsø Research Foundation via the UiT project A33020. D.D.V., T.N. and J.P.W. also

acknowledge the IS-DAAD project 255778 HOLCLIM for providing travel support. F.C.L. is partly supported by a grant from the Royal Swedish Academy of Letters, History and Antiquities and by Bank of Sweden Tercentenary Foundation (*Stiftelsen Riksbankens Jubileumsfond*). T.N. was supported by the Norwegian Research Council (KLIMAFORSK programme) under grant no. 229754. PF is supported by an NSERC-discovery grant number RGPIN-2014-05810.

This is a contribution from the interdisciplinary and international framework of the Past Global Changes (PAGES) 2k initiative (Arctic2k),

which in turn received support from the U.S. and Swiss National Science Foundations.





**Figure A1.** Calibration and Validation results. Top row: $\overline{\text{CRPS}}_{\text{pot}}$ and Reliability score for the calibration (quadrilaterals) and validation (points) period. Bottom row: CRPS scores corresponding to a ensemble-based version of the reduction of error (RE) and coefficient of efficiency (CE) estimates. Squares denote grid cells with positive CRPS-RE or CRPS-CE, indicating a skilful reconstruction in the validation period. Grid cells with few data in the validation period show a lack of skill, which might be an artifact.





**Appendix A: Calibration and Validations Statistics**

In order to estimate the skill of the reconstruction two different measures are used, the average potential continuously ranked probability score ($\overline{\text{CRPS}}_{\text{pot}}$) and the reliability (Reli) score (Hersbach, 2000; Gneiting and Raftery, 2007; Werner and Tingley, 2015; Tipton et al., 2016). The reliability analyses the accuracy of the uncertainty estimates. In principle it compares the

empirical coverage rates of uncertainty bands with their respective nominal coverage rate, e.g. a 95% confidence band should contain the target truth in 95% of the time. The $\overline{\text{CRPS}}_{\text{pot}}$ measures the accuracy of the reconstruction itself, i.e. the mismatch between the best estimate and the target. In a deterministic reconstruction it is equal to the mean absolute error. Both measures retain the original units of the data. The results are shown in Figure A1 (top row). For the calibration (validation) interval, the $\overline{\text{CRPS}}_{\text{pot}}$ is mostly below $0.1°C$ ($0.3°C$), and the Reliability is sharper than $0.1°C$. This in principle indicates a relatively low

reconstruction error, with uncertainty bands that (within reason) reflect the correct uncertainties.

Additionally the skill of the reconstruction beyond forecasting the calibration or validation period mean is evaluated. In palaeoclimate reconstructions this is often assessed by the Coefficient of Efficiency and the Reduction of Error statistics (Cook et al., 1994). However, these are not proper scoring rules (Gneiting and Raftery, 2007) and should thus not be used analysing the results of a probabilistic reconstruction method. To generate a similar statistic, the mean and standard deviation over the

validation and calibration intervals for each location with instrumental data are calculated. These are then used to generate an ensemble "reconstruction", mimicking the calibration and validation intervals. The $\overline{\text{CRPS}}_{\text{pot}}$ over the validation period for these is calculated, and then compared it against the one calculated from the actual ensemble of reconstructions. (Figure A1 bottom row). About half of the grid cells with instrumental data have a $\overline{\text{CRPS}}_{\text{pot}}$-CE and $\overline{\text{CRPS}}_{\text{pot}}$-RE that is above zero – and these grid cells are actually also those that have the longest instrumental time series (inside and outside the calibration

interval). Thus, these results not only reflect a possibly weak reconstruction but more likely the lack of actual instrumental data to construct any meaningful comparison statistics over the validation period.

**Appendix B: Intra-ensemble variance of the reconstruction**

Figure A2 presents the time changes in the spatially averaged intra-ensemble variance as a measure of the spread across the ensemble members. The variance shows a progressive decline over the pre-industrial reconstruction more pronounced in the

confidence intervals (CIs) for the period 800–1000 CE (which is linked with the time of an expansion of the multi-proxy network). Along with the intra-ensemble variability, a progressive increase in the proxy data density over time contributes to the observed decrease in the ensemble spread. The introduction of the instrumental data into the scheme (corresponding to a calibration period in the regular climate reconstruction language) causes a sharp drop in the spread after 1850 CE that reaches a minimum around 1950 CE, a period of the maximal instrumental data coverage. Figure A3 further illustrates the effects of the

spatial changes in input data density on the reconstruction intra-ensemble spread. The figure presents intra-ensemble spatial variances averaged over four time periods. The selected time-slices are associated with periods of distinctly different proxy and calibration data density: part of the Roman Warm Period 200–300 CE with a CF reconstruction based on 8 proxy records





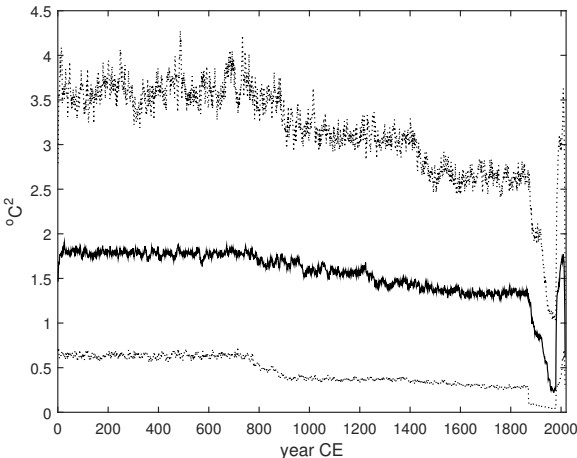

**Figure A2.** Time variability of spatially averaged intra-ensemble variance of the Arctic2k reconstruction together with the respective ensemble-based 95% CIs.

only, coldest period of the LIA 1600–1700 CE with a complete multi-proxy network, and parts of the calibration period of 1850–1900 CE and 1950–1980 CE, representative of the low and high instrumental data coverage, respectively.

## Appendix C:  Statistical properties of the reconstruction and Input Data

As an additional test for the reliability of the proxy series and the validity of the climate field reconstruction, we analyse the scaling properties of both. The used methods require the analysed time series data to be normally distributed. The Kolmogorov-Smirnov test is first used to test the normality of the proxy records and the climate field reconstruction, with a significance level $p = 0.05$. Additionally, QQ-plots are analysed. Then, the power spectral density (PSD) is used to study the variability on different time-scales for the records that were not deviating substantially from a normal distribution, using the periodogram as an estimator of the PDS. The periodogram is defined here in terms of the discrete Fourier transform $H_m$ as $S(f_m) = (2/N)|H_m|^2$, $m = 1, 2, \ldots, N/2$. The sampling time is the time unit (here: years), and the frequency is measured in cycles per time unit: $f_m = m/N$. $\Delta f = 1/N$ is the frequency resolution and the smallest frequency which can be represented in the spectrum, while $f_{N/2} = 1/2$ is the Nyquist frequency (the highest frequency that can be resolved).

The characteristic shape of the spectrum provides useful information about the temporal persistence or memory of the underlying process if the data is close to Gaussian and monofractal. If the spectrum has a power-law shape, the process exhibits long-range memory (LRM). The strength of memory in an LRM stochastic process is described by the spectral exponent $\beta$, which can be estimated by a linear fit to the power spectrum; $\log S(f) = -\beta \log f + c$. If the spectrum is Lorentzian (power law on high frequencies, flat on low frequencies), the underlying process is closer to an AR(1) process. In all spectral analyses, the fitting is applied to log-binned periodograms to ensure that all time scales are weighted equally.



**Figure A3.** Time averaged intra-ensemble variance of the Arctic2k reconstruction shown for the four subperiods with a distinct difference in proxy data density (200–300 CE vs. 1600–1700 CE, panels a and b) and calibration subperiods with different instrumental data coverage (1850–1900 vs. 1950–1980 CE, panels c and d). Black dots shows the proxy locations with a least one data point over the period of averaging.




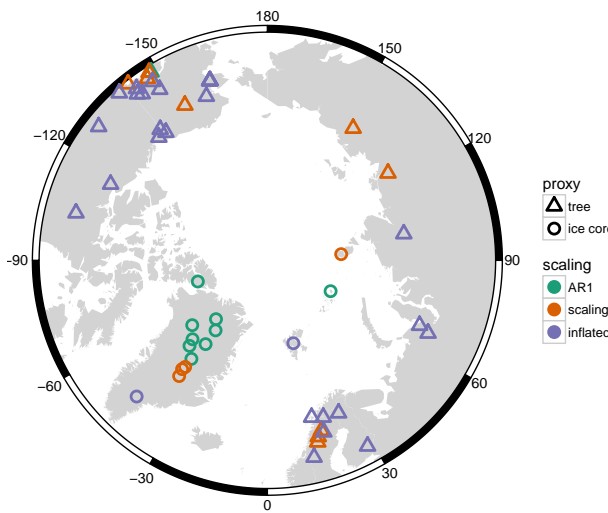

**Figure A4.** Proxy type (circle: ice cores, triangles: tree rings) and scaling properties colour coded.

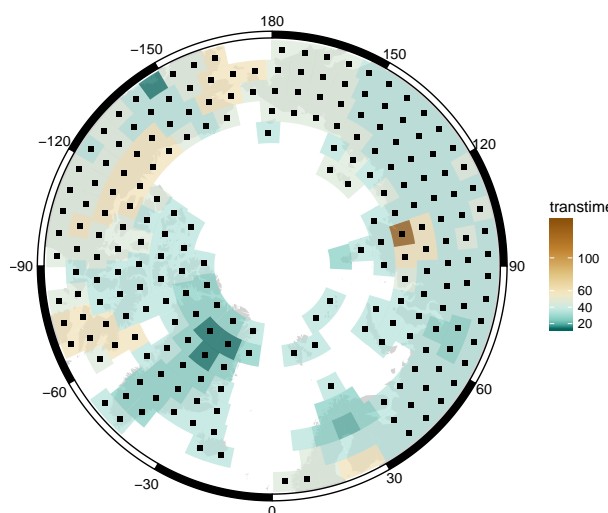

**Figure A5.** Analysis of scaling properties of the reconstruction. The transition time-scale (colour coded) and strength of change (black dot: exponent change > 1). Most of the reconstruction domain resembles an AR 1 process.



### C1   Analyses of proxy records

The six proxy records originating from lake sediments deviate substantially from a Gaussian distribution and thus had to be excluded from further spectral analyses, as second order statistics are insufficient to describe these types of records. The characteristic shape of the spectra for the remaining proxy records are then classified into three spectral categories according to their characteristics: (1) AR(1) processes, (2) persistent power law processes with spectral exponent $0 < \beta < 1$, and (3) records exhibiting weak persistence on high frequencies, and increased variability on frequencies corresponding to time scales longer than decadal–centennial. Figure A4 illustrates the spatial distribution of the proxy records with proxy type indicated by shape, and categories with colours. The Greenland records are similar to either an AR(1) or a LRM processes. The Greenland LRM records are in fact only weakly persistent, with a scaling exponent $0 < \beta < 0.3$. There is thus little evidence of long-term cooling due to orbital forcing from these records. Along with a few tree-ring records, the Greenland ice core records are the longest records used for the present reconstruction. As the low frequency variability of these records dominates the reconstructed long-term variability, the resulting reconstructions does exhibit similarly low variability at long time scales.

The proxies of category 3 are mainly tree-ring records, widely distributed along the reconstruction region. These records may require additional attention in future studies, as the level of high-versus low frequency variability is unusual compared to other proxy records and also instrumental measurements. Similar spectral characteristics were obtained for other tree-ring chronologies in (Franke et al., 2013; Zhang et al., 2015; Esper et al., 2015; Büntgen et al., 2015). The memory properties in a number of proxy-based temperature reconstructions have been studied in (Østvand et al., 2014; Nilsen et al., 2016) using the power spectrum along with selected other techniques. In these studies, LRM was detected in all records up to centennial/millennial time scales.

### C2   Analyses of climate field reconstruction

The resulting p-values from Kolmogorov-Smirnov's test indicate that for individual locations of the field reconstruction, about 80% of the ensemble members are Gaussian. For each ensemble member of the reconstruction and each location a spectral analysis is performed. Then, the ensemble-averaged spectra of each location are analysed for their scaling properties. The analyses indicates that the reconstructed temperature in all grid cells except one is best described as an AR(1) process in time. This is not surprising, as the longest proxy records exhibit low levels of long-term variability, and the BARCAST reconstruction technique assumes an AR(1) model for the temporal evolution of the temperature. Further details about the characteristic transition times are obtained by making a least-squares fit of a bilinear continuous function for the spectrum. The detected break is located where the two lines intersect.

The coloured map in FigureA5 shows the spatial distribution of the found transition time scales, black dots indicate that the difference between the scaling exponents for low and high frequencies is more than one. The spatial coherence indicates that BARCAST performs well when extrapolating temperatures to locations where observations are unavailable. For most of the area we find a marked break in the scaling (black symbols). Only the Scandinavian sector has slightly less difference between the high- and low-frequency variability, that is the scaling exponent does not change much between the two identified scaling



regimes. This indicates more similarity to a LRM process. Additionally, the transition time scale is close to century scale for

some regions, and even above a hundred years for a single location in Arctic Russia. There, the reconstruction is indeed closer

to a LRM process than an AR(1) process.

**Appendix D: Input proxy data**

**D1  Time-scale modelling**

In order to account for possible chronological uncertainties in the annually resolved proxy records, the technique of Comboul

et al. (2014) is applied to the proxies with layer counted time-scales in for the generation of ensembles of chronologies.

BAM (Banded Archive Modelling) simulates the time-scale counting procedure as a superposition of two cumulative Poisson

processes with age perturbations associated with two categories of errors either miss (type 1) or double-count (type 2) of an

annual layer. More specifically, for each measurement $x_i$ assigned a time $t_i$ with $i \in \{1, ..., n\}$, and a neighbouring $x_{i+1}$ with

$t_{i+1}$, $i \in \{2, ..., m\}$, the vector of time increments $\boldsymbol{\delta}$, $t_{i+1} - t_i = \delta_i$ comprises two independent stochastic processes $P^{\Theta_1}$ and

$P^{\Theta_1}$, with parameters $\Theta_1$ and $\Theta_2$, representing the rates of missing and doubly counted annual layers, respectively. We note

that the approach implicitly assumes the independence of the two stochastic processes and depth(time) invariance of the error

rates.

For the proxy series with chronologies constructed using a combination of annual layer counting and time markers (tie

points) $t_k$, $k \in \{1, ..., K\}$, such as volcanic sulphate peaks or tephras with ages known to a specific precision ($\sigma_k$), a two-step

procedure was implemented. The first step involved an MCMC simulation of $M$ perturbed sets of tie points $\left[t_k^{\tilde{m}}\right]$ following

Divine et al. (2012), where $[\bullet]$ stands for rounding the argument to the nearest integer. For each particular set $m$ of perturbed

tie points and a time interval $\left[t_k^{\tilde{m}}, t_{k+1}^{\tilde{m}}\right]$, $k \in \{1, ..., K-1\}$ between the perturbed pairs of tie points time-scale modelling was

applied, and only those that satisfied a criterion of $\sum \delta_i = t_k^m \tilde{} + 1 - t_k^{\tilde{m}}$ were retained for further analysis. For ages older than

$t_K^{\tilde{m}}$ a model with a free boundary was used instead. In total $M = 1000$ time-scales $\tilde{t}_i^m$ per proxy archive were generated. Using

interpolation, the proxy series $x_i$ were further projected on the generated time-scales $\tilde{t}_i^m$ to yield the ensemble of proxy series

with perturbed chronologies.

The error rates $\{\Theta_1, \Theta_2\}$ were estimated for each particular proxy archive. In the framework the counting procedure is

defined, for each point $t_i$ of the true unknown time-scale the uncertainty of the modelled time-scale follows the Skellam

distribution with parameters $\{\lambda_1, \lambda_2\} = \{(t_s - t_i)\Theta_1, (t_s - t_i)\Theta_2\}$ where $(t_s - t_i)$ denotes the time lapse between $t_i$ and a

counting start point $t_s$ (Comboul et al., 2014). For a symmetric error rate $\Theta_1 = \Theta_2$ and $(t_s - t_i)$ large enough, it converges to

a normal distribution $N(0, \lambda_1 + \lambda_2)$. The error rates can therefore be estimated as

$$\left\{\hat{\Theta}_1, \hat{\Theta}_2\right\} = \text{argmin}_{\Theta_1, \Theta_2}(\sqrt{\delta t_{max} * (\Theta_1 + \Theta_2)} - \Delta_t/4), \tag{D1}$$

where for a particular proxy archive $\delta t_{max} = \text{argmax}_k (t_{k+1} - t_k)$, $k \in \{1, ..., K-1\}$, or the entire length of the chronology,

and $\Delta_t$ denotes an estimated largest offset of the reported time-scale from the unknown true time-scale. For the majority of

records we estimated the type 1 and type 2 error rates using the authors reports on the tie point used and uncertainty of the



constructed chronologies. For the few archives where the chronological uncertainties were not reported, a conservative estimate of $[\Theta_1, \Theta_2]\, [0.05, 0.05]$ was assigned.

Table A1 shows the list of proxy series together with parameters of the model used to simulate the annual layer counting process. In total ensembles of time-scales for 13 annually dated records of the Arctic2k network, 6 ice-core and 7 lake sediment records, plus six annually dated ice cores from the Greenland German traverse from 1985 (recently reanalysed by Weißbach et al., 2016), are generated.

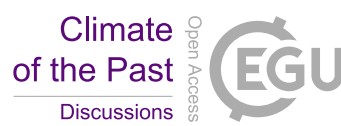



**Table A1.** List of the proxy records including the proxies of the Arctic2k network (PAGES 2k Consortium, 2017) with layer counted time-scales used in the present study together with parameters of the probabilistic model used in MC simulations of the layer counting process. The archives with a lacking information on the dating uncertainty are marked '*', a conservative estimate of $[\Theta_1,\Theta_2]$ [0.05, 0.05] was used in time-scale modelling.

| Site ID | Pages2k site name | Location Lat, Lon | Elev., m | Proxy type | Season (month) | Time period (yr CE) | $(\Theta_1,\Theta_2)$ | Reference | Data URL |
|---|---|---|---|---|---|---|---|---|---|
| Arc_025 | Lake Nautajärvi | 61.81 24.68 | 104 | ls | 3 4 5 | -555 - 1800 | 0.055 | Ojala and Alenius (2005) | https://www.ncdc.noaa.gov/paleo/study/8660 |
| Arc_076 | Soper Lake | 62.92 -69.88 | 14 | ls | 6 | 1514 - 1992 | 0.01 | Hughen et al. (2000) | https://www.ncdc.noaa.gov/paleo/study/5465 |
| Arc_024 | Donard Lake | 66.73 -61.35 | 500 | ls | 6 7 8 | 752 - 1992 | 0.05* | Moore et al. (2001) | https://www.ncdc.noaa.gov/paleo/study/6234 |
| Arc_029 | Big Round Lake | 69.87 -68.83 | 180 | ls | 7 8 9 | 971 - 2000 | 0.025 | Thomas and Briner (2009) | https://www.ncdc.noaa.gov/paleo/study/6203 |
| Arc_020 | Lake C2 | 82.13 -77.15 | 1.5 | ls | 6 7 8 | -1 - 1987 | [0.053 0.007] | Lamoureux and Bradley (1996) | https://www.ncdc.noaa.gov/paleo/study/8662 |
| Arc_004 | Lower Murray Lake | 81.35 -69.53 | 106 | ls | 7 | -1 - 2000 | 0.045 | Cook et al. (2008) | https://www.ncdc.noaa.gov/paleo/study/6195 |
| Arc_065 | Lomonosovfonna | 78.87 17.43 | 1250 | ic | 12 1 2 | 1598 - 1997 | 0.05 | Divine et al. (2011) | ://doi.pangaea.de/10.1594/PANGAEA.824732 |
| Arc_064 | ANIK | 80.52 94.82 | 750 | ic | ann | 900 - 1998 | 0.05 | Opel et al. (2013) | https://www.ncdc.noaa.gov/paleo/study/2494 |
| Arc_031 | NGRIP1 | 75.1 -42.32 | 2917 | ic | ann | -1 - 1995 | 0.0002 | Vinther et al. (2010) | https://www.ncdc.noaa.gov/paleo/study/2494 |
| Arc_034 | Dye-3 | 65.18 -43.83 | 2480 | ic | ann | 1 - 1978 | 0.0002 | Vinther et al. (2010) | https://doi.pangaea.de/10.1594/PANGAEA.786354 |
| Arc_035 | GRIP | 72.58 -37.64 | 3238 | ic | ann | 1 - 1979 | 0.0002 | Vinther et al. (2010) | https://www.ncdc.noaa.gov/paleo/study/11131 |
| Arc_032 | Agassiz Ice Cap | 80.7 -73.1 | 1700 | ic | ann | -1 - 1972 | 0.0002 | Vinther et al. (2010) | https://doi.pangaea.de/10.1594/PANGAEA.786356 |
| Arc_033 | Crête | 71.12 -37.32 | 3172 | ic | ann | 553 - 1973 | 0.0002 | Vinther et al. (2010) | https://www.ncdc.noaa.gov/paleo/study/17796 |
| Arc_011 | GISP2 | 72.10 -38.08 | 3200 | ic | ann | 818 - 1987 | 0.01 | Grootes and Stuiver (1997) | |
| Arc_078 | Windy Dome | 81.0 64.0 | 509 | ic | ann | 1225 - 1995 | 0.02 | Kinnard et al. (2011) | |
| 16 | B16 | 73.94 -37.63 | 3040 | ic | ann | 1469 - 1992 | 0.01 | Weißbach et al. (2016) | seven://doi.pangaea.de/10.1594/PANGAEA.849148 |
| 17 | B18 | 76.62 -36.40 | 2508 | ic | ann | 874 - 1992 | 0.002 | Weißbach et al. (2016) | https://doi.pangaea.de/10.1594/PANGAEA.849150 |
| 18 | B20 | 78.83 -36.50 | 2147 | ic | ann | 777 - 1993 | 0.01 | Weißbach et al. (2016) | https://doi.pangaea.de/10.1594/PANGAEA.849152 |
| 19 | B21 | 80.00 -41.14 | 2185 | ic | ann | 1373 - 1993 | 0.01 | Weißbach et al. (2016) | https://doi.pangaea.de/10.1594/PANGAEA.849153 |
| 20 | B26 | 77.25 -49.22 | 2598 | ic | ann | 1505 - 1994 | 0.005 | Weißbach et al. (2016) | https://doi.pangaea.de/10.1594/PANGAEA.8491456 |
| 21 | B29 | 76.00 -43.49 | 2874 | ic | ann | 1471 - 1994 | 0.005 | Weißbach et al. (2016) | https://doi.pangaea.de/10.1594/PANGAEA.849237 |



**Table A2.** List of the tree ring records. A few records as of submission not yet archived online, and these will be available through (PAGES 2k Consortium, 2017). Whenever latewood density was available this was favoured over other variables.

| PAGES2k ID | site Name | reference1 | lat (°) | long (°) | elevation (m) | Period (year CE) | seasonality | data URL |
|---|---|---|---|---|---|---|---|---|
| NAm_090 | Almond Butter Lower | D'Arrigo et al. (2005) | 65.2 | −162.2 | 168 | 1607 – 2002 | summer | https://www.ncdc.noaa.gov/paleo/study/3043 |
| NAm_091 | Almond Butter Upper | D'Arrigo et al. (2005) | 65.2 | −162.2 | 213 | 1406 – 2002 | summer | https://www.ncdc.noaa.gov/paleo/study/3044 |
| Arc_065 | Arjeplog | Björklund et al. (2014) | 66.3 | 18.2 | 800 – 1200 | 2010 | summer | https://www.ncdc.noaa.gov/paleo/study/14188 |
| Arc_066 | Aramaes | Björklund et al. (2012) | 65.9 | 16.1 | 500 – 1550 | 2010 | summer | https://www.ncdc.noaa.gov/paleo/study/3587 |
| NAm_002 | Avam–Taimyr | Briffa et al. (2008) | 72 | 101 | 250 | 0 – 2000 | 6 7 | https://www.ncdc.noaa.gov/paleo/study/3047 |
| NAm_096 | Big Bend Lake | Davi (2003) | 61.3 | −142.7 | 1040 | 1557 – 1997 | summer | https://www.ncdc.noaa.gov/paleo/study/3879 |
| NAm_092 | Burnt Over | D'Arrigo et al. (2005) | 65.2 | −162.3 | 259 | 1621 – 2002 | summer | https://www.ncdc.noaa.gov/paleo/study/3592 |
| NAm_089 | Canyon Creek | PAGES 2k Consortium (2017) | 63.3 | −147.8 | 884 | 1642 – 1997 | summer | https://www.ncdc.noaa.gov/paleo/study/4409 |
| NAm_126 | Coppermine River | Jacoby and D'Arrigo (1989) | 67.2 | −115.9 | 213 | 1428 – 1977 | summer | https://www.ncdc.noaa.gov/paleo/study/14790 |
| NAm_085 | Eureka Summit | Schweingruber and Briffa (1996) | 61.8 | −147.3 | 960 | 1654 – 1983 | summer | https://www.ncdc.noaa.gov/paleo/study/19743 |
| NAm_104 | Firth River | Anchukaitis et al. (2013) | 68.7 | −141.6 | 790 | 1073 – 2002 | summer | https://www.ncdc.noaa.gov/paleo/study/3264 |
| Arc_007 | Gulf of Alaska | Wiles et al. (2014) | 61.03 | −146.59 | 230 | 800 – 2010 | 2 3 4 5 6 7 8 | https://www.ncdc.noaa.gov/paleo/study/3598 |
| NAm_083 | Herring Alpine | PAGES 2k Consortium (2017) | 60.4 | −147.8 | 275 | 1422 – 1972 | summer | https://www.ncdc.noaa.gov/paleo/study/14188 |
| NAm_127 | Hornby Cabin | Jacoby and D'Arrigo (1989) | 64 | −103.9 | 160 | 1491 – 1984 | summer | https://www.ncdc.noaa.gov/paleo/study/19743 |
| Arc_016 | Indigirka | Hughes et al. (1999) | 69.5 | 147 | 80 | 1259 – 1994 | 6 7 | https://www.ncdc.noaa.gov/paleo/study/3706 |
| Arc_063 | Jämtland | Zhang et al. (2016) | 63.2475 | 13.3375 | 650 | 783 – 2011 | 4 5 6 7 8 9 | https://www.ncdc.noaa.gov/paleo/study/13785 |
| Arc_068 | Kittelfjäll | Björklund et al. (2012) | 65.2 | 15.5 | 550 | 1550 – 2007 | 6 7 | https://www.ncdc.noaa.gov/paleo/study/19851 |
| NAm_002 | Kobuk/Noatak | PAGES 2k Consortium (2017) | 67.1 | −159.6 | 100 | 978 – 1992 | summer | https://www.ncdc.noaa.gov/paleo/study/5244 |
| NAm_032 | Landslide | Clague et al. (2006) | 60.2 | −138.5 | 800 | 913 – 2001 | summer | https://www.ncdc.noaa.gov/paleo/study/3605 |
| Arc_073 | Mackenzie Delta | Porter et al. (2013) | 68.625 | −133.87 | 5 | 1245 – 2007 | 6 7 | https://www.ncdc.noaa.gov/paleo/study/13710 |
| NAm_088 | Miners Well | PAGES 2k Consortium (2017) | 60 | −141.7 | 650 | 1428 – 1995 | summer | http://www.ncdc.noaa.gov/paleo/study/1003406 |
| NAm_095 | Nabesna Mine | Davi (2003) | 62.4 | −143.1 | 1167 | 1471 – 1997 | summer | https://www.ncdc.noaa.gov/paleo/study/14274 |
| NAm_103 | Northern Alaska Composite | D'Arrigo et al. (2006) | 67 | −152 – 200 | 1524 | 1990 | summer | https://www.ncdc.noaa.gov/paleo/study/15520 |
| Eur_003 | Northern Scandinavia | Esper et al. (2012) | 68 | 25 | 300 | −138 – 2006 | summer | https://www.ncdc.noaa.gov/paleo/study/13708 |
| NAm_003 | Prince William Sound | Barclay et al. (1999) | 60.5 | −148.3 | 100 | 873 – 1991 | summer | https://www.ncdc.noaa.gov/paleo/study/2939 |
| NAm_147 | Rock Glacier Yukon | PAGES 2k Consortium (2017) | 61.4 | −128.4 | 1450 | 1697 – 2002 | annual | https://www.ncdc.noaa.gov/paleo/study/3615 |
| NAm_101 | Seward Composite | D'Arrigo et al. (2006) | 65.2 | −162.3 | 100 | 1389 – 2001 | summer | https://www.ncdc.noaa.gov/paleo/study/13709 |
| NAm_112 | Spruce Creek | Cropper and Fritts (1981) | 68.6 | −138.6 | 381 | 1570 – 1977 | summer | https://www.ncdc.noaa.gov/paleo/study/19743 |
| Arc_077 | Tjeggelvas | Björklund et al. (2012) | 66.6 | 17.6 | 520 | 1550 – 2010 | 6 7 8 | http://www.ncdc.noaa.gov/paleo/study/16790 |
| NAm_093 | Windy Ridge Alaska | D'Arrigo et al. (2005) | 65.2 | −162.2 | 251 | 1556 – 2002 | summer | https://www.ncdc.noaa.gov/paleo/study/3902 |
| NAm_102 | Wrangells Composite | D'Arrigo et al. (2006) | 62 | −142 | 200 | 1540 – 1998 | summer | https://crudata.uea.ac.uk/cru/papers/melvin2012h |
| Arc_061 | Polar Urals | Schneider et al. (2015) | 66.9 | 65.6 | 250 | 891 – 2006 | 6 7 8 | https://crudata.uea.ac.uk/cru/papers/briffa2013qs |
| Eur_013 | Finnish Lakelands | Helama et al. (2014) | 62 | 28.325 | 130 | 760 – 2000 | summer | https://www.ncdc.noaa.gov/paleo/study/13758 |
| Arc_074 | Forfjorddalen | McCarroll et al. (2013) | 68.73 | 15.73 | 200 | 1100 – 2007 | 6 7 | |
| Arc_071 | Laanila | Lindholm et al. (2011) | 65.0952 | 63.4091 | 270 | 745 – 2007 | 6 7 8 | |
| Arc_024 | Lower Lena River | MacDonald et al. (1998) | 70.67 | 125.87 | 180 | 1490 – 1994 | 6 | |
| Arc_062 | Torneträsk | Melvin et al. (2013) | 68.26 | 19.6 | 320 | −39 – 2010 | 6 7 8 | |
| Arc_079 | Yamalia | Briffa et al. (2013) | 68.4 | 66.85 | 30 | 914 – 2003 | 6 7 | |
| Arc_008 | Yukon | D'Arrigo et al. (2006) | 67.9 | −140.7 | 300 | 1177 – 2000 | summer | |





**Table A3.** List of the data from the PAGES2k database that was not used in this study.

| Site ID | Pages2k site name | Location Lat, Lon | Proxy type | Time period (yr CE) | Resolution | Reason |
|---|---|---|---|---|---|---|
| Arc_037 | Iceland | 64.77,-18.37 | doc | 945 – 1935 | 30 | not annual |
| Arc_005 | Camp Century | 77.17,-61.13 | ic | 1242 – 1967 | 1 | 20 yr averages |
| Arc_018 | Austfonna | 79.83,24.02 | ic | 1400 – 1998 | 1 | interpolated onto annual scale |
| Arc_075 | Prince-of-Wales, Ellesmere Isl. | 78.4,-80.4 | ic | 151 – 1995 | 1 | not annual as per original article description |
| Arc_044 | Devon Ice Cap | 75.33,-82.5 | ic | 1 – 1971 | 5 | not annual |
| Arc_059 | Renland | 71.27,-26.73 | ic | 3 – 1983 | 5 | not annual |
| Arc_045 | Penny Ice Cap P96 | 67.25,-66.75 | ic | 5 – 1980 | 25 | not annual |
| Arc_001 | Blue Lake | 68.09,-150.47 | ls | 730 – 2000 | 1 | very nonlinear response, short overlap with instrumental, unclear interpretation |
| Arc_014 | Lake Lehmilampi | 63.62,29.1 | ls | 1 – 1800 | 1 | exact interpretation unclear from original article |
| Arc_022 | Hvítárvatn | 64.6,-19.8 | ls | -1 – 2000 | 1 | annual and centennial signal inconsistent |
| Arc_069 | Kongressvatnet | 78.0217,13.9311 | ls | 232 – 2008 | 10 | not annual |
| Arc_067 | Hallet Lake | 61.5,-146.2 | ls | 116 – 2005 | 11 | not annual |
| Arc_050 | Lake Hamptriäsk | 60.28,25.42 | ls | 1359 – 1994 | 14 | not annual |
| Arc_070 | Lake E | 67,-50.7 | ls | -3642 – 1876 | 19 | not annual |
| Arc_043 | Braya Sø | 67,-50.7 | ls | -998 – 1999 | 29 | not annual |
| Arc_040 | Moose Lake | 61.35,-143.6 | ls | -718 – 1963 | 36 | not annual |
| Arc_051 | Lake Pieni-Kauro | 64.28,30.12 | ls | 462 – 1979 | 44 | not annual |
| Arc_041 | Hudson Lake | 61.9,-145.66 | ls | -837 – 1997 | 47 | not annual |
| Arc_054 | Lake 4 | 65.1,-83.79 | ls | 634 – 1997 | 50 | not annual |
| Arc_042 | Screaming Lynx Lake | 66.07,-145.4 | ls | -1067 – 1988 | 51 | not annual |





**Appendix: Data availability**

The input proxy data is available either through the individual publications (see tables), the majority will also be available bundled through the latest publication of the PAGES 2k Consortium (2017) unless where indicated otherwise (NGT ice cores).

The used data will also be available as the BARCAST input data (see below).

The base instrumental data (see Harris et al., 2014) can be downloaded from the BADC, the most recent version can be reached from the CRU homepage `http://browse.ceda.ac.uk/browse/badc/cru/data/cru_ts/cru_ts_3.24.01` under "observations".

The treated input data and the R script files used for the treatment of the input data as well as the reconstruction results
(ensemble reconstruction, gridded ensemble mean and area mean) will be made available through NOAA / WDC when the final paper is published. The Fortran and R code based on Werner and Tingley (2015) will be made available through BitBucket. Contact JPW for obtaining the code beforehand.



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
