# Peer review of "Spatio-temporal variability of Arctic summer temperatures over the past two millennia"

_Climate of the Past, 2017_

## Referee Comment (RC1) · Anonymous Referee #1 · 17 Apr 2017

The authors present a new Arctic summer temperature reconstruction over the past 2,00 years using a Bayesian reconstruction technique. The paper discusses and thoroughly analyzes this new Arctic temperature reconstruction. The method and analysis seem sound and consistent. I just have one large concern in reading the manuscript: I didn't get a sense that it was addressing any particular scientific question and it was hard for me to see where the paper was going scientifically. I think the authors have produced an important and valuable reconstruction, but I think having a more specific direction or question to address would make the paper much more interesting and useful. It's not obvious, to me at least, that a paper devoted to "an overview of the last major climate anomalies" would garner a lot of interest, especially when it is primarily

about one particular reconstruction. Perhaps the paper might be better focused around an issue like their findings that contemporary warming doesn't conclusively supersede the MCA peak? There's plenty of material here to tell an interesting story, but I don't think any coherent story is being told. The authors do have some interesting analysis discussing the finding I just suggested that could be used to focus the body of the text as well as focus the somewhat vague title. I would recommend (though not absolutely insist) that the authors focus the paper on one or two specific scientific/climate questions.

Minor points:

Abstract, lines 1-2: I think you need to be a little more clear about what is actually unique here. I understand that it is BOTH spatially resolved AND millennial in length, though there are several reconstructions that are one or the other.

Paragraph including lines 53-60: This paragraph comes across as a kind of special attack on the glacier advances work in a tone that I'm not sure the authors intended. This summer temperature reconstruction (with skill primarily over Europe) is really different than a glacial reconstruction given the memory of glaciers, the different seasons and climate factors a glacier is responding to, etc. So I don't think a clear declaration against that work is necessarily warranted.

Line 72: define LOC

Lines 124-125: Need a more specific criticism here or not discuss the issue at all. What constitutes a "strange" correlation? And on what firm basis can you reject the use of the BE product?

Lines 147, 444: Question mark issues.

Lines 199-202: How are the response parameters being determined? Do your reconstructions happen to take account of the specific choice of parameter values?

Lines 211-222: I think it's important to note that only Europe has spatially coherent

skill, otherwise it's fairly patchy skill (at least in my reading of Fig. A1).

Figure A1: It wasn't clear to me what is meant by "CRPS CE" and "CRPS RE"? As a related issue, CRPS is challenging to interpret because it doesn't have a reference. Perhaps use the skill score version of CRPS that takes account of a reference (e.g., your prior)?

Fig 2b: Why is there so much precision right up to the end of the calibration interval, but a complete loss of annual precision from 1980 to the present? Are important proxies dropping out here?

Figure 5: no color on the color labels

Lines 458-459 "which gets sparser going back in time"

Line 5:14 "used for these chronologies"

---

## Referee Comment (RC2) · N. McKay (Referee) · 2 May 2017

Werner et al., present a new climate field reconstruction for the terrestrial Arctic (>60N) based on 54 annually-resolved temperature sensitive record. The reconstruction is created using an extended version of the age-uncertain BARCAST methodology presented in Werner and Tingley, 2015, which enables the authors to reconstruct the climate field while accounting for age uncertainty in the uncertain layer counted records (from ice cores and varved sediment). The result is a probabilistic CFR that extends back to 750 AD (although this is inconsistent in the manuscript – see below) and a reconstructed Arctic mean that extends through the Common Era. This represents a major advance, both scientifically, as this is the longest and most data-rich CFR yet developed for the Arctic, and methodologically, as this is the first full-fledged CFR that I'm aware of that rigorously incorporates age uncertainty. These are both major scientific publications that warrant publication in Climate of the Past.

The authors explore the new reconstruction by examining the 1) predominant trends in the CFR and index reconstruction, 2) reconstructed decades and centuries of extreme warm and cold conditions, and their relative distinctness and 3) evidence for arctic amplification in this reconstruction and a similar reconstruction created for Europe.

The manuscript is also generally well written, methodologies and assumptions are well articulated and the scientific importance of the results is generally well handled. The figures are also of professional quality, although I do include comments where I believe they could be improved in several instances.

Despite the high-impact science presented here, and the professional presentation, I have a number of concerns that I believed must be addressed before publication. My primary concern is the remarkable finding by the authors of widespread and significant warming trends throughout Greenland and Eastern Canada. These are presented clearly in figure 4a. First – this is an instance where the temporal extent of the CFR is confusing, as the trend map is presented as linear trends from 1-1850 CE, not 750-1850 CE. The larger concern, however, is that the origin of these warming trends is mysterious, in that they're not supported by the data. I've attached maps of the trends in annually resolved records north of 60N in the PAGES 2k v2.0.0 database, for records that extend from 1, 750, and 1000 CE to 1850 CE. Although, as the authors note, the trends in the datasets are relatively weak comparative to the interannual variability, most of the records demonstrate significant cooling trends over those intervals, and the rest show insignificant cooling trends. There essentially no evidence in the datasets for warming trends in Greenland and Eastern Greenland.

Given this – this result in the CFR is particularly interesting, and I'm keen to understand why this is occurring. I wonder about the possibility that in fitting the model parameters

described on line 182 to each record based on the instrumental data, that the direction of the linear proxy-climate relationship is inverted in the Greenland records relative to how the data were interpreted by the original authors. Regardless of whether or not this is the cause of the warming trends, including the key parameters in tables A1 and A2 would be helpful to the reader. Also regardless, this CFR/data discrepancy and its cause must be discussed by the authors. To this end, I suggest that the trends of the proxy records be added to figure 4a to make this comparison clear.

Until this concern is resolved, it's hard to evaluate the significance of the epoch analysis and arctic-wide trend comparisons, as its possible that they might change. For example, if the discrepancy in Greenland trends is resolved, the overall cooling trend in the reconstruction will likely increase towards significance, and towards previous estimates.

Regarding the third major scientific topic – that of Arctic observation, I have some bigger picture questions. Primarily – given the methodologies used here, where the parameters that scale proxy data to climate are fit for each record (in the Arctic) or record type (in Europe) relative to their fit with instrumental data – is it possible to learn anything substantial about Arctic amplification in the past? In other words, doesn't the parameter fitting essentially force the apparent amplitude of Arctic change to be greater than lower latitude change because the same phenomenon is observed in the instrumental data, which are used to estimate the temperature scaling for each record? Maybe I've missed something, but my suspicion is that random data run the this approach would also reveal Arctic Amplification (AA) just do to the way the parameters in the model are estimated.

Despite the argument above, there may be good evidence that I am misunderstanding something here given that the Arctic data in Figure 8 do not follow this pattern; although the European data do (increase variability with latitude) as expected. My understanding is that this is due to the inability to directly compare the Arctic and European results, although I aslo don't understand why they cannot be compared directly. If this is indeed

the case, I'd appreciate a much fuller understanding of why they cannot be compared.

Also in this section is the discussion of lagged response of peak (and minimum) temperature in the Arctic relative to Europe, presented in Figure 9. I think it's hazardous to make this assertion, as I don't think there's any evidence that the Arctic warming observed ca. 1000 AD is at all related to the warming in Europe ca. 950 AD, especially given that the European warm intervals ca. 750 and 1200 CE have no Arctic counterpart, and there's no other evidence that medieval climate in the two regions are connected.

Ultimately, given the apparent inability to compare the European and Arctic reconstructions, the potential challenges in looking at AA in such reconstructions, and the challenges in relating temperature variability between the two regions, I'm not sure that this section belongs in the manuscript, because as it currently stands, it raises more questions than it answers.

I suggest that the authors replace this with a different investigation with less complications. I like the idea of comparing the Arctic reconstruction with the European one (but only if they can be directly compared) but limiting the focus to the areas of overlap, or near overlap, would lead to a more interesting discussion, as there the assumption that they should be covarying is much more reasonable.

Overall, this is an exciting study and a well-written paper. After the issues discussed above are satisfactorily resolved, I suggest that it be published in Climate of the Past.

I attached detailed comments in a marked version of the manuscript.

Sincerely, Nick McKay

Please also note the supplement to this comment:
http://www.clim-past-discuss.net/cp-2017-29/cp-2017-29-RC2-supplement.pdf

[Figure]

Normalized.PAGES2kv2.0.0.annual.trends.1-1850

[Figure]

**Fig. 1.**

Normalized.PAGES2kv2.0.0.annual.trends.750-1850

[Figure]

**Fig. 2.**

[Figure]

Normalized.PAGES2kv2.0.0.annual.trends.1000-1850

**Fig. 3.**

**Supplement:**

[revised manuscript text omitted]

---

## Editor Comment (EC1) · D.s Kaufman (Editor) · 22 May 2017

The PAGES Data Stewardship Integrative Activity seeks to advance best practices for sharing data generated and assembled as part of all PAGES-related activities. As part of this activity, a team of reviewers has been constituted for the "Climate of the Past 2000 years" Special Issue. The data team is reviewing the data handling within each of the CP-Discussion papers in relation to the CP data policy and current best practices. The team has identified essential and recommended additions for each paper, with the goal of achieving a high and consistent level of data stewardship across the 2k Special Issue. We recognize that an additional effort will likely be required to meet the high level of data stewardship envisaged, and we appreciate the dedication

and contribution of the authors. This includes the use of Data Citations (see example in supplement). We ask authors to respond to our comments as part of the regular open interactive discussion. If you have any questions about PAGES Data Stewardship principles, please contact any of us directly.

Best wishes for the success of your paper,

2k Special Issue Data Review Team (Darrell Kaufman, Nerilie Abram, Belen Martrat, Raphael Neukom, Scott St. George) and ex-officio team members (Marie-France Loutre, Lucien von Gunten)

Essential additions for this paper:

(1) Move the "Data Availability" section from the Appendix to the main text and update it to include the Data Citation/URL for the reconstructions generated in this study.

(2) Add the missing "data URL" to Tables A1 and A2. Note the typos in a few of the URLs as listed.

(3) Correct the Site IDs in Table A1 so they match the PAGES IDs in version 2.0 of the PAGES 2k database (e.g., Lake Nautajarvi is Arc_026, not Arc_025 as listed in the table). Change "Site ID" to "PAGES2k ID"

(4) Submit the primary outcome of this study (ensemble reconstruction, gridded ensemble mean and area mean) to a public repository and include the Data Citation/URL in "Data Availability".

Recommended element:

(5) Add a link to the BitBucket URL that includes the code used for the reconstruction.

Please also note the supplement to this comment:
http://www.clim-past-discuss.net/cp-2017-29/cp-2017-29-EC1-supplement.pdf

---

## Author Comment (AC1) · 13 Jun 2017

Essential additions for this paper:

(1) Move the "Data Availability" section from the Appendix to the main text and update it to include the Data Citation/URL for the reconstructions generated in this study.

OK. (see also 4, 5)

(2) Add the missing "data URL" to Tables A1 and A2. Note the typos in a few of the URLs as listed.

(3) Correct the Site IDs in Table A1 so they match the PAGES IDs in version 2.0 of

the PAGES 2k database (e.g., Lake Nautajarvi is Arc_026, not Arc_025 as listed in the table). Change "Site ID" to "PAGES2k ID"

as noted in the submission phase (editor comments) we will update the reconstruction input data to version 2 of the data base. We will carefully update the proxy data tables.

(4) Submit the primary outcome of this study (ensemble reconstruction, gridded ensemble mean and area mean) to a public repository and include the Data Citation/URL in "Data Availability".

Recommended element:

(5) Add a link to the BitBucket URL that includes the code used for the reconstruction

These two points are also in our own interest: Data and code that are not accessible will not get used and thus not cited. The data and code will be hosted at NOAA or NPI.

---

## Author Comment (AC2) · 21 Jun 2017

**General Comment**

(paraphrased) No scientific question explicitly posed, article lacking focus.

Reply:

We will remove the Arctic Amplification chapter, which will result already in a more coherent manuscript, making connections between the different chapters much clearer.

[Figure]

After revising the reconstruction we will also try and make stronger statements about e.g. the spatial consistency of the warming and cooling episodes.

**Other comments**

- Abstract, lines 1-2: I think you need to be a little more clear about what is actually unique here. I understand that it is BOTH spatially resolved AND millennial in length, though there are several reconstructions that are one or the other.

R: we will strengthen that statement in the final version of the paper and stress that it is both the spatial character and temporal aspect (as guessed correctly)

- Paragraph including lines 53-60: This paragraph comes across as a kind of special attack on the glacier advances work in a tone that I'm not sure the authors intended. This summer temperature reconstruction (with skill primarily over Europe) is really different than a glacial reconstruction given the memory of glaciers, the different seasons and climate factors a glacier is responding to, etc. So I don't think a clear declaration against that work is necessarily warranted.

R: We apologize for the perceived tone. We did not mean to criticize glacial reconstructions from moraines in general and cosmogenic dating in particular. The sentence will be removed as it is also based on a misinterpretation of the results by Young et al. (pers. comm. from Young set this straight).

- define LOC

R: it stands for "local regression", though mostly when the method is referred to in other articles the abbreviation LOC seems to be used. Still, we will change this.

- Lines 124-125: Need a more specific criticism here or not discuss the issue at all. What constitutes a "strange" correlation? And on what firm basis can you reject the use of the BE product?

R: the issue is that the correlation between grid cells as a function of the distance (both chordal and orthodromic) is very long (10 000 km) and contains oscillatory parts (not as obvious in the attached figure as in other evaluations). Without analysing any details of the regridding method used in the BEST data, this looks too much like an artefact of an expansion in spherical functions. Truncating after a certain order can lead to spurious oscillations on the sphere. See the attached figure.

- Lines 147, 444: Question mark issues.

R: missing reference in the bibTeX file. Will be fixed

- Lines 199-202: How are the response parameters being determined? Do your reconstructions happen to take account of the specific choice of parameter values?

R: The parameters are estimated using the described Gibbs sampler. The reconstructions are conditional on the estimated joint distribution of all parameters (proxy response and climate field). That is, it explicitly takes the uncertainty in the parameters into account.

- Lines 211-222: I think it's important to note that only Europe has spatially coherent skill, otherwise it's fairly patchy skill (at least in my reading of Fig. A1).

R: The skill shown in Figure A1 depends on the length (and quality) of the instrumental data in the specific grid cell. The data coverage over Europe is highest (space and time), other regions (absence of colour in Fig. A1) are less well covered (thus the patchy appearance). Any estimates relying on short time series

are thus to be interpreted carefully. This is what we mean by "Thus, these results not only reflect a possibly weak reconstruction but more likely the lack of actual instrumental data to construct any meaningful comparison statistics over the validation period." (l. 550-551)

- Figure A1: It wasn't clear to me what is meant by "CRPS CE" and "CRPS RE"? As a related issue, CRPS is challenging to interpret because it doesn't have a reference. Perhaps use the skill score version of CRPS that takes account of a reference (e.g., your prior)?

R: We did indeed skim over this issue. We will modify the last two paragraphs accordingly: "Additionally the skill of the reconstruction beyond forecasting the calibration or validation period mean is evaluated. In palaeoclimate reconstructions this is often assessed by the Coefficient of Efficiency and the Reduction of Error statistics (Cook et al., 1994). However, these are not proper scoring rules (Gneiting and Raftery, 2007) and should thus not be used analysing the results of a probabilistic reconstruction method.

To generate a similar statistic, ensembles of surrogates for each location with instrumental data are constructed using the mean and standard deviation over the validation interval from the reconstructions. For these, the $CRPS_{pot}$ is calculated, comparing the surrogates against the instrumental target. This value is then subtracted from the $CRPS_{pot}$ over the calibration (validation) interval, resulting in $CRPS_{pot}$-CE ($CRPS_{pot}$-RE). As with the CE (RE) a value above zero shows a skilful reconstruction, i.e. a reconstruction that performs better than the climatology over the calibration (validation) interval.

About half of the grid cells with instrumental data have a $CRPS_{pot}$-CE and $CRPS_{pot}$-RE that is above zero – and these grid cells are actually also those that have the longest instrumental time series (inside and outside the calibration interval). Thus, these results not only reflect a possibly weak reconstruction

but more likely the lack of actual instrumental data to construct any meaningful comparison statistics over the validation period."

- Fig 2b: Why is there so much precision right up to the end of the calibration interval, but a complete loss of annual precision from 1980 to the present? Are important proxies dropping out here?

R: The precision appears indeed relatively low, and Fig.2b greatly emphasises this over Fig.2c, especially since we have the "calibration" interval so prominently in there. One issue is that "calibration" is not the entirely correct term, as the reconstruction over this period is in fact mostly determined by the instrumental data (though not entirely, not the relatively high instrumental uncertainty and the spatially sparse coverage). This issue can be addressed by doing what is called a predictive run (see Luterbacher et al. 2016, or Tingley and Huybers 2013), which would in turn give another means of evaluating the reconstruction quality. However, as can be seen from 2d, this is also very likely a proxy availability effect.

- Figure 5: no color on the color labels

R: These were present in the initially uploaded pdf, we will look into the technical issues behind that. Most likely once the original artwork is uploaded and the LaTeX process takes place at Copernicus this issue will disappear.

**Typos / grammatical issues:**

- Lines 458-459 "which gets sparser going back in time"

- Line 5:14 "used for these chronologies"

Thanks for catching these!

Figure 1: Correlation between the gridded instrumental series of July SAT of the BE dataset as a function of the orthodromic distance between the grid nodes. Nodes located within 45-90N were used in the analysis.

[Figure]

[Figure]

**Fig. 1.**

---

## Author Comment (AC3) · 21 Jun 2017

We are greatful for the very detailed reveiw and the extremely thorough commenting in the uploaded pdf! In the following we will try to first answer the most important (as we understood them) comments and take the rest in shorter form later.

**1 Warming trend**

1. The trend map from 1–1850 CE. The gridded reconstruction itself is (repeatedly mentioned!) only going back to 750 CE.

2. The warming trend over Greenland is not supported by the raw proxy data

Reply:

1. True, although the assessment this is based on relies on the spread of the annual reconstruction of the past. The trend should in principle be more robust against this. We propose the following solution: Only plot trends back to 1 CE in regions where the reconstruction is estimated to retain skill. Additionally plot a trend analysis going back only to 750 CE, though of course this could be influenced by the onset of the MCA.

2. We will also add the trends of the individual proxy series on the map. The one presented by the reviewer did contain a few that we discarded, but it is a step in the right direction.

3. Even more important is it to show a map with an estimated SNR (or the plain $\beta_1$, the scaling of the individual proxies) – as also commented by the reviewer. See the attached figure for the current estimates (bound to be revised in the updated reconstruction). The numbers will be added either in the proxy data tables in the appendix or in an own table, depending on the typographical limitations. (see attached figure)

**2 Arctic Amplification**

The comparability of reconstructions is noted as questionable (or at least difficult) in the text, and thus the exercise is criticised (rightly so). There also seemed to be a misunderstanding regarding the analysis, in the sense that we limit ourselves to the European sector of the arctic – the circum-polar behaviour being indeed quite different timing and amplitude wise. We have decided to remove this chapter entirely, and wait for comprehensive reconstructions based on the North American and the Asian data.

**3 Lake data**

We made indeed a linear response assumption for the lake data, which might be defended by noting that the interpretation of some of the used records is based on the cross correlation to instrumental data, i.e. the linear assumption. However, we have now decided to transform the data using the inverse quantile transformation. It will be interesting to see if the data is then weighted more in the actual reconstruction (preliminary results suggest so, though not overly much).

**4 Other comments**

The reviewer caught several typos and a few strange (i.e. wrong) grammatical constructs, these will (of course) be fixed.

- Fig 7, flip axes? add data coverage

R: We will add the data coverage. We will also try to toy with the axes orientation, however we feel that the current orientation with the longitude in the "natural"

orientation is superior to a time axis going left to right. We also updated the estimate of what constitutes a "significant" warm or cold event.

- C1: how is this screenign used?

R: This is a misunderstanding, we mostly did this to see how the reconstruction changes the LRM properties. It has been commented by others (also at meetings and conferences) that basing the reconstruciton on an explicit spatial and temporal model is bound to change the correlation (in space and time) behaviour of the resulting reconstruction (see e.g. Raible et al. Clim.Chage 2006 for the effect on the spatial correlations in EOF based reconstructions). In principle, all spatio-temporal reconstruction mehtods impose an explicit model. In the classical world (PCA, CCA, . . . ), the spatial patterns are truncated and the temporal process is assumed to be i.i.d. In other methods (Ed Cook's PPR), the data is pre-whitened to remove auto-correlations first, and then do a regression, while basing the spatial correlations on the instrumental period and imposing a convex spatial structure through the search radius. This will be clarified in the final version of the manuscript

[Figure]

**Fig. 1.**

---

## Editor Comment (EC2) · D.s Kaufman (Editor) · 30 Jun 2017

Dear Authors: Thank you for your replies to the reviewers' comments. In my opinion, your responses are sufficient for a decision on the publication – except for one remaining critical issue. Referee #2 (McKay) pointed out that there is no evidence in the proxy dataset that supports the preindustrial warming trend in Greenland and eastern Canada, as is shown in your reconstruction. He suspected that the direction of the proxy relations might have been reversed in your formulation. I agree with the referee that this important disagreement and its cause must be diagnosed and discussed before the paper can proceed. Therefore, please prepare a response for this

interactive discussion that includes an explanation of the discrepancy between the observational/proxy data and the reconstruction. If there is an error in the reconstruction, please take the time to correct it and explain how the revised reconstruction will influence the primary conclusions of the paper.

Sincerely, Darrell Kaufman

---

## Author Comment (AC4) · 8 Nov 2017

As commented by Nick McKay, the trends over Greenland (1-1850 CE) contradict other evidence. He suggested that some of the proxy records (the ice core data) was likely flipped during the reconstruction. We found that indeed, the ice core and lake sediment data were flipped, though not in sign but in direction (ordering in terms of age or year CE). As this was consistently done with all of the ice cores, the trend over Greenland was essentially inverted in time (also pers. comm. to the editor).

We have now fixed this issue, see the attached figure. The trend is colour-coded, with the proxies going back to at least the first century CE marked on the map. We will

address the changes in the revised version of the paper.

Remark: While the proxy network before about 750CE is too sparse to get a meaningful reconstruction over the whole domain, the reconstruction at or close to the proxy locations (within about 1500 km) should still be skillful (see the methodological papers by Tingely and Huybers 2010, or the paper of Werner et al. 2013). This will be discussed (and visualised better) in the revisions.

We are grateful for getting the chance of revising the reconstruction accordingly, and for the patience of the editors (article and SI) and reviewers.

For the authors, Johannes Werner

————————————————————

[Figure]

[Figure]

**Fig. 1.** Trends over the first 1850 years of the reconstruction. Proxy sites with data going back into the first century CE are marked with dots.

---

## Author Response (AR1)

[Figure]

Figure 1: Correlation between two grid cells vs. their distance in the BEST data

We have extensively rewritten the article: The chapter about the Arctic Amplification was removed. Instead we have now included a more in-depth discussion (and analysis) of the trends over the Common Era.

As noted in one of our Discussion replies, there was an error in how the lake and ice core data was imported into BARCAST. This resulted in the strange trend pattern over Greenland. We identified the issue now, and the reconstruction was updated accordingly. Thus, many changes in the article were neccesary, on top of those pointed out by the reviewers.

We are grateful for the patience of both the editors in charge and the reviewers, and hope that this revised manuscript can now be considered for publication. Below follows the point-by-point reply to the issues raised during the review.

**1 Reply to Reviewer 1**

**General Comment**

(paraphrased) No scientific question explicitly posed, article lacking focus.

**Reply:** We will remove the Arctic Amplification chapter, which will result already in a more coherent manuscript, making connections between the different chapters much clearer. After revising the reconstruction we will also try and make stronger statements about e.g. the spatial consistency of the warming and cooling episodes.

**Other comments**

- Abstract, lines 1-2: I think you need to be a little more clear about what is actually unique here. I understand that it is BOTH spatially resolved AND millennial in length, though there are several reconstructions that are one or the other.

R: we changed that statement in the final version of the paper and stress that it is both the spatial character and temporal aspect (as guessed correctly)

- Paragraph including lines 53-60: This paragraph comes across as a kind of special attack on the glacier advances work in a tone that Im not sure the authors intended. This summer temperature reconstruction (with skill primarily over Europe) is really different than a glacial reconstruction given the memory of glaciers, the different seasons and climate factors a glacier is responding to, etc. So I dont think a clear declaration against that work is necessarily warranted.

R: We apologize for the perceived tone. We did not mean to criticize glacial reconstructions from moraines in general and cosmogenic dating in particular. The sentence will be removed as it is also based on a misinterpretation of the results by Young et al. (pers. comm. from Young set this straight).

- define LOC

R: it stands for "local regression"

- Lines 124-125: Need a more specific criticism here or not discuss the issue at all. What constitutes a strange correlation? And on what firm basis can you reject the use of the BE product?

R: the issue is that the correlation between grid cells as a function of the distance (both chordal and orthodromic) is very long (10 000 km) and contains oscillatory parts (not as obvious in the attached figure as in other evaluations). Without analysing any details of the regridding method used in the BEST data, this looks too much like an artefact of an expansion in spherical functions. Truncating after a certain order can lead to spurious oscillations on the sphere. See the attached figure.

- Lines 147, 444: Question mark issues.

R: missing reference in the bibTeX file. fixed

- Lines 199-202: How are the response parameters being determined? Do your reconstructions happen to take account of the specific choice of parameter values?

R: The parameters are estimated using the described Gibbs sampler. The reconstructions are conditional on the estimated joint distribution of all parameters (proxy response and climate field). That is, it explicitly takes the uncertainty in the parameters into account.

- Lines 211-222: I think its important to note that only Europe has spatially coherent skill, otherwise its fairly patchy skill (at least in my reading of Fig. A1).

R: The skill shown in Figure A1 depends on the length (and quality) of the instrumental data in the specific grid cell. The data coverage over Europe is highest (space

and time), other regions (absence of colour in Fig. A1) are less well covered (thus the patchy appearance). Any estimates relying on short time series are thus to be interpreted carefully. This is what we mean by "Thus, these results not only reflect a possibly weak reconstruction but more likely the lack of actual instrumental data to construct any meaningful comparison statistics over the validation period." (l. 550-551)

- Figure A1: It wasnt clear to me what is meant by CRPS CE and CRPS RE? As a related issue, CRPS is challenging to interpret because it doesnt have a reference. Perhaps use the skill score version of CRPS that takes account of a reference (e.g., your prior)?

R: We did indeed skim over this issue. We modified the last two paragraphs accordingly, and (hopefully) made the reasoning behind this much clearer.

- Fig 2b: Why is there so much precision right up to the end of the calibration interval, but a complete loss of annual precision from 1980 to the present? Are important proxies dropping out here?

R: The precision appears indeed relatively low, and Fig.2b greatly emphasises this over Fig.2c, especially since we have the "calibration" interval so prominently in there. One issue is that "calibration" is not the entirely correct term, as the reconstruction over this period is in fact mostly determined by the instrumental data (though not entirely, not the relatively high instrumental uncertainty and the spatially sparse coverage). This issue can be addressed by doing what is called a predictive run (see Luterbacher et al. 2016, or Tingley and Huybers 2013), which would in turn give another means of evaluating the reconstruction quality. However, as can be seen from 2d, this is also very likely a proxy availability effect.

- Figure 5: no color on the color labels

R: These were present in the initially uploaded pdf, we will look into the technical issues behind that. Most likely once the original artwork is uploaded and the LaTeX process takes place at Copernicus this issue will disappear.

**Typos / grammatical issues:**

- Lines 458-459 which gets sparser going back in time

- Line 5:14 used for these chronologies

Thanks for catching these!

We are greatful for the very detailed reveiw and the extremely thorough commenting in the uploaded pdf! In the following we will try to first answer the most important (as we understood them) comments and take the rest in shorter form later.

**2 Warming trend**

1. The trend map from 1–1850 CE. The gridded reconstruction itself is (repeatedly mentioned!) only going back to 750 CE.

2. The warming trend over Greenland is not supported by the raw proxy data

1. True, although the assessment this is based on relies on the spread of the annual reconstruction in the past. The trend should in principle be more robust against this. We did additionally plot a trend analysis going back only to 750 CE, though of course this could be influenced by the onset of the MCA.

2. Even more important is it to show a map with an estimated SNR (or the plain $\beta_1$, the scaling of the individual proxies) – as also commented by the reviewer. See Fig. A6 for the current estimates.

**3 Arctic Amplification**

The comparability of reconstructions is noted as questionable (or at least difficult) in the text, and thus the exercise is criticised (rightly so). There also seemed to be a misunderstanding regarding the analysis, in the sense that we limit ourselves to the European sector of the arctic – the circum-polar behaviour being indeed quite different timing and amplitude wise. We have decided to remove this chapter entirely, and wait for comprehensive reconstructions based on the North American and the Asian data.

**4 Lake data**

We made indeed a linear response assumption for the lake data, which might be defended by noting that the interpretation of some of the used records is based on the cross correlation to instrumental data, i.e. the linear assumption. However, we have now decided to transform the data using the inverse quantile transformation.

**5 Other comments**

The reviewer caught several typos and a few strange (i.e. wrong) grammatical constructs, these will (of course) be fixed.

- Fig 7, flip axes? add data coverage

R: We will add the data coverage. We will also try to toy with the axes orientation, however we feel that the current orientation with the longitude in the "natural" orientation is superior to a time axis going left to right. We also updated the estimate of what constitutes a "significant" warm or cold event.

- C1: how is this screenign used?

R:  This is a misunderstanding, we mostly did this to see how the reconstruction changes the LRM properties. It has been commented by others (also at meetings and conferences) that basing the reconstruciton on an explicit spatial and temporal model is bound to change the correlation (in space and time) behaviour of the resulting reconstruction (see e.g. Raible et al. Clim.Chage 2006 for the effect on the spatial correlations in EOF based reconstructions). In principle, all spatio-temporal reconstruction mehtods impose an explicit model. In the classical world (PCA, CCA, ...), the spatial patterns are truncated and the temporal process is assumed to be i.i.d. In other methods (Ed Cook's PPR), the data is pre-whitened to remove auto-correlations first, and then do a regression, while basing the spatial correlations on the instrumental period and imposing a convex spatial structure through the search radius.

**Spatio-temporal variability of Arctic summer temperatures over the past two millennia: an overview of the last major climate anomalies**

Johannes P. Werner[1], Dmitry V. Divine[2,3], Fredrik Charpentier Ljungqvist[4,5], Tine Nilsen[3], and Pierre Francus[6,7]

[1]Bjerknes Centre for Climate Research and Department for Earth Science, University of Bergen, PO Box 7803, N-5020 Bergen, Norway
[2]Norwegian Polar Institute, FRAM Centre, N-9296 Tromsø, Norway.
[3]Department of Mathematics and Statistics, University of Tromsø – The Arctic University of Norway, N-9037, Norway
[4]Department of History, Stockholm University, SE-106 91 Stockholm, Sweden
[5]Bolin Centre for Climate Research, Stockholm University, SE-106 91 Stockholm, Sweden
[6]Centre - Eau Terre Environnement, Institut National de la Recherche Scientifique, 490 rue de la couronne, Québec, QC G1K 9A9, Canada
[7]GEOTOP Research Center, Montréal, H3C 3P8, Canada

*Correspondence to:* J.P. Werner (johannes.werner@geo.uib.no)

**Abstract.**

In this article, the first spatially resolved millennium-long and millennium-length summer (June–August) temperature reconstruction over the Arctic and Subarctic domain (north of 60° N) is presented. It is based on a set of 54 44 annually dated temperature sensitive proxy archives of various types, mainly from the updated and revised PAGES2k database supplemented

5  with 6 new recently published updated proxy records. As a major noveltyadvance, an extension of the Bayesian BARCAST climate field (CF) reconstruction technique provides a means to treat climate archives with dating uncertainties. In total 1400 over 600 independent realisations of the temperature CF were generated, enabling further analyses to be carried out in a probabilistic framework. The new seasonal CF reconstruction for the Arctic region can be shown to be skilful for the majority of the terrestrial nodes. The decrease in the proxy data density back in time however limits the analyses in the spatial domain to

10  the period after 750 CE, while the spatially averaged reconstruction covers the entire time interval of 1–2002 CE. The analysis of basic features of the reconstructed seasonal CF focuses on the regional expression of past major climate anomalies in order to uncover the potential of the new product for studying Common Era temperature variability in the region.

The long-term, centennial to millennial, evolution of the reconstructed temperature is in good agreement with a general pattern that was inferred in recent studies for the Arctic and its sub-regions. The reconstruction shows a pronounced Me-

15  dieval Climate Anomaly (MCA, here, ca. 960–1060920–1060 CE), which was characterised by a sequence of extremely warm decades over the whole domain. The medieval warming was followed by a gradual cooling into the Little Ice Age (LIA), with 1580–16801766–1865 CE as the longest centennial-scale cold period, culminating around 1812–18221811–1820 CE for most of the target region. At the same time there is evidence for a drastic reduction in sea-ice on the Greenland shelf, which is reflected by rather high summer temperatures over Greenland and Baffin Island during that decade.

20     While our analysis shows that the peak MCA summer temperatures were as high as in the late 20th and early 21st century, the spatial coherence of extreme years over the last decades of the reconstruction (1980s onwards) seems unprecedented at least back until 750 CE. However, statistical testing could not

25    provide conclusive support of the contemporary warming to  exceed the peak of the MCA in terms of the pan-Arctic mean summer temperatures: neither can the reconstruction be extended reliably past 2002 CE due to lack of proxy data and thus the most recent warming is not captured, nor is it (from a statistical viewpoint) advisable to directly compare the reconstruction and instrumental data.

[revised manuscript text omitted]

870 The two largest statistically significant cooling rates in the entire ensemble with average temperature changes of $-0.05 \pm 0.01$°C/year and $-0.04 \pm 0.01$°C/year over three decades are registered at 1450 CE and 1669 CE, respectively, while a recovery after the first cooling centered at 1477 CE featured a warming rate of $0.04 \pm 0.01$°C/year over the same time period. In terms of the rate of changes attained, the first cooling/warming episode appears unique over the 2000-year long reconstruction, embracing one of the coldest decades in the reconstruction ensemble. At the highlighted centennial timescale, the most rapid changes are

875 the MCA to LIA transition with a cooling of $-0.006 \pm 0.002$°C/year centered at 1040 CE, cooling towards one of the LIA SAT minima at 1577 CE with $-0.04 \pm 0.02$°C/year, and the transition to CWP centered at 1905 CE with an average warming rate of $0.01 \pm 0.001$°C over about 30 years.

**Appendix: Data availability**

The

**Table A3.** List of the lake and ice core data from the PAGES2k database that was not used in this study.

[revised manuscript text omitted]

---

## Referee Report (RR1)

**Spatio-temporal variability of Arctic summer temperatures over the past two millennia**

Johannes P. Werner[1], Dmitry V. Divine[2,3], Fredrik Charpentier Ljungqvist[4,5], Tine Nilsen[3], and Pierre Francus[6,7]

[1]Bjerknes Centre for Climate Research and Department for Earth Science, University of Bergen, PO Box 7803, N-5020 Bergen, Norway
[2]Norwegian Polar Institute, FRAM Centre, N-9296 Tromsø, Norway.
[3]Department of Mathematics and Statistics, University of Tromsø – The Arctic University of Norway, N-9037, Norway
[4]Department of History, Stockholm University, SE-106 91 Stockholm, Sweden
[5]Bolin Centre for Climate Research, Stockholm University, SE-106 91 Stockholm, Sweden
[6]Centre - Eau Terre Environnement, Institut National de la Recherche Scientifique, 490 rue de la couronne, Québec, QC G1K 9A9, Canada
[7]GEOTOP Research Center, Montréal, H3C 3P8, Canada

*Correspondence to:* J.P. Werner (johannes.werner@geo.uib.no)

**Abstract.**

   In this article, the first spatially resolved and millennium-length summer (June–August) temperature reconstruction over the Arctic and Subarctic domain (north of 60° N) is presented. It is based on a set of 44 annually dated temperature sensitive proxy archives of various types, mainly from the updated and revised PAGES2k database supplemented with 6 new recently updated proxy records. As a major advance, an extension of the Bayesian BARCAST climate field (CF) reconstruction technique provides a means to treat climate archives with dating uncertainties. In total over 600 independent realisations of the temperature CF were generated, enabling further analyses to be carried out in a probabilistic framework. The new seasonal CF reconstruction for the Arctic region can be shown to be skilful for the majority of the terrestrial nodes. The decrease in the proxy data density back in time however limits the analyses in the spatial domain to the period after 750 CE, while the spatially averaged reconstruction covers the entire time interval of 1–2002 CE. The analysis of basic features of the reconstructed seasonal CF focuses on the regional expression of past major climate anomalies in order to uncover the potential of the new product for studying Common Era temperature variability in the region.

   The long-term, centennial to millennial, evolution of the reconstructed temperature is in good agreement with a general pattern that was inferred in recent studies for the Arctic and its sub-regions. The reconstruction shows a pronounced Medieval Climate Anomaly (MCA, here, ca. 920–1060 CE), which was characterised by a sequence of extremely warm decades over the whole domain. The medieval warming was followed by a gradual cooling into the Little Ice Age (LIA), with 1766–1865 CE as the longest centennial-scale cold period, culminating around 1811–1820 CE for most of the target region.

   While our analysis shows that the peak MCA summer temperatures were as high as in the late 20th and early 21st century, the spatial coherence of extreme years over the last decades of the reconstruction (1980s onwards) seems unprecedented at least back until 750 CE. However, statistical testing could not provide conclusive support of the contemporary warming to exceed

the peak of the MCA in terms of the pan-Arctic mean summer temperatures: neither can the reconstruction be extended reliably past 2002 CE due to lack of proxy data and thus the most recent warming is not captured, nor is it (from a statistical viewpoint) advisable to directly compare the reconstruction and instrumental data.

[revised manuscript text omitted]

---

## Author Response (AR2)

**To the editior**

Dear Hugues Goosse,

attached are the point-by-point replies to the reviewers' comments, as well as the marked-up version of the article highlighting the changes. We hope that we have adequately addressed their questions and comments, and hope that this study can now be accepted for publication.

Best,

Johannes Werner (for the authors)

**Reviewer 1, anonymous**

We thank the anonymous reviewer for their input! Below follows the point by point reply.

**Remark 1**

> However, I think my main criticism from the first round of reviews still stands in that the article is definitely not hypothesis driven. Specifically, the article is not framed around answering a scientific question or hypothesis, rather its framed around the production and presentation of a reconstruction product. Its true that the authors do many specific and quantitative analyses of the reconstruction and these do constitute assessing many hypotheses. But none of these analyses are moving the narrative of the article along a specific line of inquiry.
>
> Perhaps my criticism here is off-base in that the presentation of a product is perfectly acceptable for publication in Climate of the Past. Theres nothing wrong with the science as far as I can see. But even if this article just remains primarily about the presentation of a product, I think it needs some more indications of the types of questions that one could answer with it. Why should I as a reader and potential user of the product get excited about the fact that it exists? What question(s) can I answer with it that I couldnt have answered before? What benefit would I get in using a product like this that, more so than any other approach I think, explicitly accounts for many different types of uncertainty? More emphasis of reader-interest concerns like this could definitely be included in the abstract and conclusions without much more work for the authors.

We don't entirely agree with a criticism of Reviewer 1 regarding the lacking scientific hypothesis behind the presentation. Even if the study may appear to be too focused on the methodological aspects of making a new reconstruction, resolving these methodological issues common within the topic of multiproxy-based paleoclimate studies actually represents a scientific problem in itself. Throughout the manuscript we hope we have demonstrated that the proposed solution is scientifically sound and presented in a consistent way (as the reviewer does seem to agree with). Furthermore, we emphasized the benefits of employing the introduced novelties for addressing a number of questions that are difficult to solve in a conventional way.

The authors are also grateful for the questions the Reviewer has formulated in the second part of the above comment. It actually motivated and helped us to modify the abstract, introduction and conclusions. This was done in the way that, in our opinion, makes the paper to appear more oriented towards a reader primarily interested in a climate-related output of the study.

We have thus modified the abstract, the introduction and the conclusions to highlight the type of questions that can be answered in a novel way by using the output of this

reconstruction. The reviewer is right in assuming that there are more things to get excited about than just having yet another reconstruction, as nice as it seems to be. The discussion was rewritten to highlight this by stating:

> As highlighted in Sec. 4, the probabilistic nature of the reconstruction results in straightforward uncertainty estimates even for complex analyses. As quantiles for a particular type of analysis are evaluated for individual ensemble members, the overall intra-ensemble coherence determines the spread and hence uncertainty of these quantities. The resulting ensemble of reconstructions including the ensemble of likely chronologies thus provides a convenient dataset for further studies.

**Remark 2**

> One minor question. It is not clear to me what the authors mean in the last sentence of the abstract: "nor is it (from a statistical viewpoint) advisable to directly compare the reconstruction and instrumental data." This seems mis-phrased somehow. Doesnt every reconstruction including this one verify their reconstructions against observation/instrumental data? And shouldnt we expect that a reconstruction behave at least a little bit like instrumental products?

Yes, in principle the reconstructions should behave like the instrumental data - at least they should be closely related to them. However, using the instrumental data with very different spatio-temporal availability in a direct comparison with the reconstruction is not entirely like-for-like. One could of course "splice" the instrumental data to the reconstruction (BARCAST does this implicitly under certain conditions), but the problems do remain. One issue is for example whether the proxy response is saturated wrt. to warm temperatures (such as the "divergence" problem that was discussed for tree ring series, which might be an expression of the loss of limiting factors as described by Tolwinski-Ward in her VS-Lite articles). We have briefly tested subsampling an infilled instrumental data grid at the proxy locations, and it indeed closely matches the reconstruction (outside the calibration period) and shows even warmer temperatures but this would be quite ad-hoc and would not address the proxy response problem.

We tried to address this question carefully by mentioning that the last decade was indeed warmer than the previous decades in the instrumental data, and thus could have exceeded the MCA temperatures.

**Reviewer 2, N. McKay**

We are grateful for the very detailed comments that Nick McKay provided in both rounds of reviews (also in the commented pfd file). This has improved the article greatly from the first submission. In the following, we offer a point-by-point reply to the last issues raised.

Q: One item that I continue to be confused about is the temporal duration of the reconstruction (much to the authors frustration, I believe). The authors say that the gridded reconstruction is limited to the time after 750 CE, and otherwise imply that the field reconstruction begins in 750, however other times they simply say that the analysis is limited to the time after 750 CE, and they also show reconstructed trends, spatially, for the period from 1-1850 CE in figure 6a. After twice reviewing the manuscript, and two responses from the authors, my current suspicion is that spatial reconstruction covers the period from 1-2002, however is not robust before 750, and they urge caution when interpreting it, only examining it themselves when looking at long term trends (and urging caution). This is fine, it just needs to be well (and consistently explained). Also, if Im correct, will the full 1-2002 interval be included in the reconstructed data fields? My opinion is that they should be, with caution urged anywhere the data are hosted.

A: We understand the question (and your frustration), and we hope that we have addressed this in most places. Your intuition is, of course, right: The area average is (as we note) the area average of each ensemble member, which we then proceed to analyse. Before about 750 CE, the reconstruction is no longer skilful over much of the reconstruction region (due to proxy availability). We had mentioned the truncation e.g. in the Reconstruction Quality section, where we explicitly state that

> This analysis hints that while there could still be skill left in the mean Arctic summer temperature reconstruction in the first centuries CE, the precision of the spatial reconstruction rapidly decreases in areas that become more data sparse. While the reconstruction over the regions with local proxy data present – such as Fennoscandia – remains reliable, a time-varying reconstruction domain (or rather, domain over which the reconstruction is analysed) would be beyond the scope of this paper. Thus the gridded reconstruction is only shown back to 750 CE. However, for single analyses over data rich regions the full reconstruction period (1–2002 CE) can in principle be used.

We have revised the manuscript where we mention the limitation to make this more obvious and more consistent throughout.

Q 1. Data: I was glad to see the data availability section said that all the input data and code would be available  does this mean the input proxy data, including age ensembles, as well as the instrumental target developed as part of the paper will be

available? The link is not currently active, but I hope this is the case as those data, as well as the full ensemble output, are extremely valuable for the community.

A: Yes. Everything (code, input, full ensemble output with reconstruction and parameters) will be available at NOAA.

Q 2. Figure axis labels are inconsistently handled, in terms of the orientation, and the common variable (units) structure. Several of the axis labels have what code names rather than common names.

A: We hope we have fixed this everywhere (also the missing panel marking). The "code names" likely refer to the bandwidth of the trend analysis. We have discussed this among the authors, and while it is difficult to really map a bandwidth to an explicit time scale we have changed the axis accordingly.

Q 3. The definition of the MCA used several places in the manuscript (9201060 CE) is narrower than normal, to coincide with the warmest 140 years in the reconstruction. Additionally, the authors refer to a late Roman Warm Period in the 4th and 5th centuries. Id encourage the authors to not feel obliged to identify the observed warm periods with these more classical and general names, and explicitly discuss the difference between the observed warm periods and the more general concepts.

A: It was not in our intentions to try redefining the MCA, but rather emphasise that we deal with a terrestrial circum-Arctic expression of this phenomenon. Moreover, in Section 4 we demonstrate that this expression has regional features, in line with a present day knowledge of the MCA as a spatially not entirely coherent phenomenon, especially with respect to the duration. Over Europe, for example, the MCA lasts likely until the 12$^{\text{th}}$ century, though the warm period is punctuated by colder episodes (Luterbacher et al., Env.Res.Lett. 2016).

Q 4. The abstract says 44 records, and the conclusion says 54 records.

A: Our mistake. The first version did use 54 records, for the new one we removed all tree ring series that are shorter than 500 years.

**Spatio-temporal variability of Arctic summer temperatures over the past two millennia**

Johannes P. Werner[1], Dmitry V. Divine[2,3], Fredrik Charpentier Ljungqvist[4,5], Tine Nilsen[3], and Pierre Francus[6,7]

[1]Bjerknes Centre for Climate Research and Department for Earth Science, University of Bergen, PO Box 7803, N-5020 Bergen, Norway
[2]Norwegian Polar Institute, FRAM Centre, N-9296 Tromsø, Norway.
[3]Department of Mathematics and Statistics, University of Tromsø – The Arctic University of Norway, N-9037, Norway
[4]Department of History, Stockholm University, SE-106 91 Stockholm, Sweden
[5]Bolin Centre for Climate Research, Stockholm University, SE-106 91 Stockholm, Sweden
[6]Centre - Eau Terre Environnement, Institut National de la Recherche Scientifique, 490 rue de la couronne, Québec, QC G1K 9A9, Canada
[7]GEOTOP Research Center, Montréal, H3C 3P8, Canada

*Correspondence to:* J.P. Werner (johannes.werner@geo.uib.no)

**Abstract.**

In this article, the first spatially resolved and millennium-length summer (June–August) temperature reconstruction over the Arctic and Subarctic domain (north of 60° N) is presented. It is based on a set of 44 annually dated temperature sensitive proxy archives of various types  from the revised PAGES2k database supplemented with 6 new recently updated proxy records. As a major advance, an extension of the Bayesian BARCAST climate field (CF) reconstruction technique provides a means to treat climate archives with dating uncertainties. This results not only in a more precise reconstruction but additionally enables joint probabilistic constraints to be imposed on the chronologies of the used archives. The new seasonal CF reconstruction for the Arctic region can be shown to be skilful for the majority of the terrestrial nodes. The decrease in the proxy data density back in time however limits the analyses in the spatial domain to the period after 750 CE, while the spatially averaged reconstruction covers the entire time interval of 1–2002 CE.

The  centennial to millennial  evolution of the reconstructed temperature is in good agreement with a general pattern that was inferred in recent studies for the Arctic and its sub-regions.  In particular, the reconstruction shows a pronounced Medieval Climate Anomaly (MCA, here, ca. 920–1060 CE), which was characterised by a sequence of extremely warm decades over the whole domain. The medieval warming was followed by a gradual cooling into the Little Ice Age (LIA), with 1766–1865 CE as the longest centennial-scale cold period, culminating around 1811–1820 CE for most of the target region.

20    In total over 600 independent realisations of the temperature CF were generated. As showcased for local and regional trends and temperature anomalies, operating in a probabilistic framework directly results in comprehensive uncertainty estimates, even for complex analyses. For the presented multiscale trend analysis, for example, the spread in different paths across the reconstruction ensemble prevents a robust analysis of features at timescales shorter than ca. 30 years. For the spatial reconstruction, the benefit of using the spatially resolved reconstruction ensemble is demonstrated by focusing on the regional

25    expression of the recent warming and the MCA. While our analysis shows that the peak MCA summer temperatures were as high as in the late 20th and early 21st century, the spatial coherence of extreme years over the last decades of the reconstruction (1980s onwards) seems unprecedented at least back until 750 CE. However, statistical testing could not provide conclusive support of the contemporary warming to exceed the peak of the MCA in terms of the pan-Arctic mean summer temperatures: neither can the reconstruction the reconstruction cannot be extended reliably past 2002 CE due to lack of proxy data and thus

30    the most recent warming is not captured, nor is it (from a statistical viewpoint) advisable to directly compare the reconstruction and instrumental data.

[revised manuscript text omitted]